# Deep Actor-Critics with Tight Risk Certificates

## Abstract

Deep actor-critic algorithms have reached a level where they influence everyday life. They are a driving force behind continual improvement of large language models through user feedback. However, their deployment in physical systems is not yet widely adopted, mainly because no validation scheme fully quantifies their risk of malfunction. We demonstrate that it is possible to develop tight risk certificates for deep actor-critic algorithms that predict generalization performance from validation-time observations. Our key insight centers on the effectiveness of minimal evaluation data. A small feasible set of evaluation roll-outs collected from a pretrained policy suffices to produce accurate risk certificates when combined with a simple adaptation of PAC-Bayes theory. Specifically, we adopt a recently introduced recursive PAC-Bayes approach, which splits validation data into portions and recursively builds PAC-Bayes bounds on the excess loss of each portion's predictor, using the predictor from the previous portion as a data-informed prior. Our empirical results across multiple locomotion tasks, actor-critic methods, and policy expertise levels demonstrate risk certificates tight enough to be considered for practical use.

## 1 Introduction

Reinforcement learning (RL) is transforming emerging AI technologies. Large language models incorporate human feedback via RL, thereby continually improving accuracy (Christiano et al., 2017; Ziegler et al., 2019; DeepSeek-AI et al., 2025). Generative AI is increasingly integrated into agentic workflows to automate complex decision-making tasks. RL has shown great promise in controlling physical robotic systems. Recent deep actor-critic algorithms learned to make a legged robot walk after only 20 minutes of outdoor training in an online mode (Kostrikov et al., 2023). Model-based extensions of actor-critic pipelines can also achieve sample-efficient visual-control tasks in diverse settings (Hafner et al., 2025; Zhang et al., 2023). Despite promising results observed in experimental conditions, RL is used far less than classical approaches in physical robot control. This opportunity has largely been missed because deep RL algorithms are overly sensitive to initial conditions and can change behavior drastically during training. Embodied intelligent systems pose a high risk of causing harm when generalization performance differs significantly from observed validation performance. Predictable generalization performance is even more critical when these systems update their behavior based on interactions with humans.

There has been an effort to use learning-theoretic approaches to train high-capacity predictors with risk certificates, i.e., bounds that guarantee a predictor's generalization performance. Typically, this performance is estimated from observed validation results, which may be misleading. *Probably Approximately Correct Bayesian (PAC-Bayes) theory* (McAllester, 1999; Alquier et al., 2024) provides risk certificates for stochastic predictors relative to a prior distribution over the hypothesis space. In this framework, the computationally prohibitive capacity term is reduced to a Kullback-Leibler divergence between the posterior and the prior, enabling the incorporation of domain knowledge into the analysis. Since stochastic policies are often considered, relying on PAC-Bayes is a natural choice.

PAC-Bayes is the first and remains the most promising method for providing meaningful risk certificates for deep neural networks (Dziugaite & Roy, 2017; Pérez-Ortiz et al., 2021; Lotfi et al., 2022). Further studies have improved the tightness (i.e., precision) of these certificates through the following techniques: (i) pretraining probabilistic neural networks on held-out data and using them as *data-informed priors* (Ambroladze et al.,

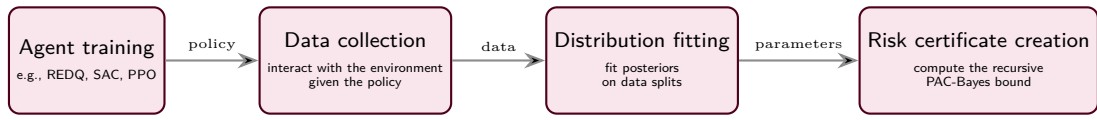

Figure 1: The four-step procedure to generate tight risk certificates for deep actor-critic algorithms.

2006; Dziugaite et al., 2021); (ii) using pretrained networks as first-step predictors and developing PAC-Bayes guarantees on the residual of their predictions, termed the *excess loss*; and (iii) recursively repeating the first two steps on multiple data splits, a recent method known as *Recursive PAC-Bayes* (Wu et al., 2024). The scope of these developments has thus far been limited to simple classification tasks with feedforward neural networks. Their application to deep actor–critic algorithms remains open, primarily because mainstream PAC-Bayes bounds assume i.i.d. datasets, whereas RL assumes a controlled Markov chain.

We present a simple recipe for providing risk certificates for deep, model-free actor–critic architectures. We find that, contrary to expectations, the three modern PAC-Bayesian learning techniques mentioned above can successfully handle the high variance of Monte Carlo samples collected by running a pretrained policy network for multiple episodes in evaluation mode. Our approach proposes self-certified training of probabilistic neural networks on different splits of an i.i.d. dataset containing return realizations of the policy, computed by first-visit Monte Carlo and post-processed through a simple thinning approach. We recursively build a PAC-Bayes bound on the excess losses of these networks, deriving a new adaptation of Wu et al. (2024). Figure 1 illustrates our risk-certificate generation workflow. We evaluate the approach for several classical actor–critic agents, PPO (Schulman et al., 2017), SAC (Haarnoja et al., 2018a), and REDQ (Chen et al., 2021), across three benchmark suites. Our results indicate that the risk certificates become tighter as the recursion depth increases. The final bounds are sufficiently tight for practical use. Furthermore, the tightness of the risk certificates is proportional to the policy's expertise.

## 2 Background

### 2.1 The state of the art of model-free deep actor-critic learning

Consider a set of states $\mathcal{S}$ an agent may be in and an action space $\mathcal{A}$ from which the agent can choose actions to interact with its environment. Denote by $\Delta(\mathcal{S})$ and $\Delta(\mathcal{A})$ the sets of probability distributions defined on $\mathcal{S}$ and $\mathcal{A}$, respectively. We define a Markov Decision Process (MDP) (Puterman, 2014) as the tuple $M = \langle \mathcal{S}, \mathcal{A}, r, P, P_0, \gamma \rangle$, where $r : \mathcal{S} \times \mathcal{A} \to [0, R]$ is a bounded reward function, $P : \mathcal{S} \times \mathcal{A} \times \mathcal{S} \to [0, 1]$ is the state-transition kernel conditioned on a state-action pair; specifically $P(s'|s, a)$ is the probability distribution of the next state $s' \in \mathcal{S}$ given the current state-action pair $(s, a) \in \mathcal{S} \times \mathcal{A}$. We denote the initial-state distribution by $P_0 \in \Delta(\mathcal{S})$, the discount factor by $\gamma \in (0, 1)$, and let $\pi : \mathcal{S} \times \mathcal{A} \to [0, 1]$ be a policy. The goal of RL is to learn a policy that maximizes the expected discounted return, $\pi_* := \arg\max_{\pi \in \Pi} \mathbb{E}_{\tau_\pi} \left[ \sum_{t=0}^{\infty} \gamma^t r(s_t, a_t) \right]$. The expectation is taken with respect to the trajectory $\tau_\pi := (s_0, a_0, s_1, a_1, s_2, a_2, \ldots)$ of states and actions generated when a policy $\pi$ chosen from a feasible set $\Pi$ is executed. We refer to $\pi_*$ as the optimal policy. The exact Bellman operator for a policy $\pi$ is defined as

$$T_\pi Q(s, a) := r(s, a) + \gamma \mathbb{E}_{s' \sim P(\cdot|s,a)} \left[ Q(s', \pi(s')) \right] \tag{1}$$

for some function $Q : \mathcal{S} \times \mathcal{A} \to \mathbb{R}$. The unique fixed point of this operator is the true action-value function $Q_\pi$, which maps a state-action pair $(s, a)$ to the expected discounted sum of rewards the policy $\pi$ collects when executed from $(s, a)$. In other words, the equality $T_\pi Q(s, a) = Q(s, a)$ holds if and only if $Q(s, a) = Q_\pi(s, a), \forall (s, a)$. Any other $Q$ incurs an error $(T_\pi Q(s, a) - Q(s, a))^2$, called the *Bellman error*. Common deep actor-critic methods approximate the true action-value function $Q_\pi$ by one-step Temporal Difference (TD) learning that minimizes $L(Q, \pi) := \mathbb{E}_{s \sim P_\pi}[(T_\pi Q(s, a) - Q(s, a))^2]$ with respect to $Q$, given a data set $\mathcal{D}$ and $P_\pi(s' \in A) = \mathbb{E}_{s \sim P_0} \left[ \sum_{t>0} P(s_t \in A | s_0 = s, \pi(s)) \right]$ which is defined as the state-visitation distribution of policy $\pi$ for some event $A$ that belongs to the $\sigma$-algebra of the transition probability distribution. Because the transition probabilities are unknown, the expectation term in (1) cannot be computed. Instead, the observed transitions are used to approximate it with a single-sample Monte Carlo estimate, yielding the

training objective below:

$$\widetilde{L}(Q) := \mathbb{E}_{s \sim P_\pi} \left[ \mathbb{E}_{s' \sim P(\cdot|s,\pi(s))} \left[ (r(s,a) + \gamma Q(s', \pi(s')) - Q(s,a))^2 \right] \right].$$

A deep actor-critic algorithm fits a neural-network function approximator $Q$, referred to as the critic, to a set of observed tuples $(s, a, s')$ stored in a replay buffer $\mathcal{D}$ by minimizing an empirical estimate of the stochastic loss: $\widehat{L}_\mathcal{D}(Q) := 1/|\mathcal{D}| \sum_{(s,a,s')\in\mathcal{D}} (\widetilde{T}_\pi Q(s, a, s') - Q(s, a))^2$. The critic is subsequently used to train a policy network, or actor, $\pi' \leftarrow \arg\max_\pi \mathbb{E}_{s \sim P_\pi}[Q(s, \pi(s))]$. It is common practice to adopt the *Maximum-Entropy Reinforcement Learning* approach (Haarnoja et al., 2018a;b) to balance exploration and exploitation, ensuring effective training. The approach supplements the reward function $r(s, a)$ with a policy-entropy term $\mathbb{H}[\pi(\cdot|s)]$, scaled by a hyperparameter $\alpha \geq 0$, tuned jointly with the actor and critic, i.e., $r_{\mathrm{MaxEnt}} \triangleq r(s, a) + \alpha \mathbb{H}[\pi(\cdot|s)]$.

Performing off-policy TD learning with deep neural nets is notoriously unstable which is often attributed to the *deadly triad* (Sutton & Barto, 2018). The main source of instability is the accumulation of errors from approximating $T_\pi Q$ by its Monte Carlo estimate. Strategies to improve stability include maintaining Polyak-updated target networks (Lillicrap et al., 2016) and learning twin critics while using the minimum of their target-network outputs in Bellman target calculation (Fujimoto et al., 2018). Empirically, training an ensemble of critic networks in a maximum-entropy setup largely mitigates these stability issues. We adopt PPO (Schulman et al., 2017), SAC (Haarnoja et al., 2018a), and REDQ (Chen et al., 2021), three popular actor-critic methods for model-free continuous control, as our representative approaches. This choice is pragmatic rather than restrictive allowing us to trade the computational cost of a broader exploration of algorithms for a deeper, more comprehensive empirical evaluation of a single one.

## 2.2 Developing risk certificates with PAC-Bayes bounds

PAC-Bayes (McAllester, 1999; Alquier et al., 2024) offers a powerful framework for understanding and controlling the generalization performance of learning algorithms by integrating prior beliefs with information obtained from data. *PAC-Bayesian learning* employs modern machine learning techniques to model $\rho$ with complex function approximators and fit them to data. It has been successfully applied to both image classification (Dziugaite & Roy, 2017; Wu et al., 2024) and regression tasks (Reeb et al., 2018). Its application to reinforcement learning has thus far been limited to the design of critic training losses, without rigorous quantification of the tightness of the performance guarantees (Tasdighi et al., 2024; 2025).

**Notation.** Let $\mathcal{H} : \mathcal{X} \to \mathcal{Y}$ be a set of feasible hypotheses, and let $\ell : \mathcal{Y} \times \mathcal{Y} \to [0, 1]$ be a bounded loss function.[1] Further, let $L(h) = \mathbb{E}_{(x,y)\sim P_D}[\ell(h(x), y)]$ denote the expected error, where $P_D$ is a distribution on $\mathcal{X} \times \mathcal{Y}$. The empirical loss is $\hat{L}(h) = \frac{1}{N} \sum_{i=1}^N \ell(h(x_i), y_i)$ for a data set $\mathcal{D} = \{(x_n, y_n) : n \in \{1, \ldots, N\}\}$ of size $N$ with $(x_n, y_n) \sim P_D$. $\mathcal{P}$ denotes the set of distributions on $\mathcal{H}$. For two distributions $\rho, \rho_0$ on $\mathcal{H}$, the Kullback–Leibler (KL) divergence is defined as $\mathrm{KL}(\rho \parallel \rho_0) \triangleq \mathbb{E}_{h\sim\rho}[\log \rho(h) - \log \rho_0(h)]$. We use $\mathrm{kl}(p \parallel q) \triangleq p \log(p/q) + (1-p) \log((1-p)/(1-q))$ to denote the KL divergence between two Bernoulli distributions. We define the upper inverse of $\mathrm{kl}(\cdot \parallel \cdot)$ as $\mathrm{kl}^{-1,+}(\hat{p}, \varepsilon) \triangleq \max\{p : p \in [0, 1] \mid \mathrm{kl}(\hat{p} \parallel p) \leq \varepsilon\}$ and the lower one as $\mathrm{kl}^{-1,-}(\hat{p}, \varepsilon) \triangleq \min\{p : p \in [0, 1] \mid \mathrm{kl}(\hat{p} \parallel p) \leq \varepsilon\}$. PAC-Bayesian analysis (McAllester, 1999; Shawe-Taylor & Williamson, 1997) develops bounds on the *expected loss* $\mathbb{E}_{h\sim\rho}[L(h)]$ under a posterior distribution $\rho$ with respect to a prior distribution $\rho_0$ that hold with high probability. That is, they provide *risk certificates* for the generalization error. For brevity, we use $\mathbb{E}_\rho[\cdot] = \mathbb{E}_{h\sim\rho}[\cdot]$ throughout this paper. In the context of PAC-Bayes, the terms *posterior* and *prior* refer to distributions dependent on and independent of the validation data, respectively. They are not to be interpreted in a Bayesian sense as being linked by a likelihood.[2] The choice of bounds that yields the tightest risk certificates depends on the specific use case; see, e.g., Alquier et al. (2024) for a recent introduction and a survey of various PAC-Bayesian bounds. Here, we rely on bounds derived from the kl divergence, as they are tighter than alternatives when no additional information about the data distribution is available, while noting that the same arguments apply to any other PAC-Bayesian bound.

---

[1]Our discussion generalizes directly to any bounded loss within an interval $[a, b]$ with $a, b \in \mathbb{R}$.

[2]See Germain et al. (2016) for results linking PAC-Bayes and Bayesian inference.

### 2.2.1 PAC-Bayes-split-kl bound

Wu & Seldin (2022) introduce a PAC-Bayes bound for random variables that take values in intervals $[a, b]$ by splitting them into components that individually satisfy the constraints of a kl-inequality (see Lemma A.2).

Define $\tilde{\ell} : \mathcal{Y} \times \mathcal{Y} \to [a, b]$, where $a, b \in \mathbb{R}$. For $\mu \in [a, b]$, define $\tilde{\ell}^+ = \max\{0, \tilde{\ell} - \mu\}$ and $\tilde{\ell}^- = \max\{0, \mu - \tilde{\ell}\}$. $\tilde{L}^+(h) = \mathbb{E}_{(x,y) \sim P_D}[\tilde{\ell}^+(h(x), y)]$ and $\hat{\tilde{L}}^+(h) = \frac{1}{N} \sum_{n=1}^{N} \tilde{\ell}^+(h(x_n), y_n)$ are the expected and empirical losses. $L^-$ and $\hat{\tilde{L}}^-$ are defined analogously. We cite the following bound.

**Theorem 2.1** (PAC-Bayes-Split-kl inequality (Wu & Seldin, 2022)). *Let $\tilde{\ell}$ and the remaining loss terms be defined as above. Then, for any $\rho_0$ on $\mathcal{H}$ independent of $\mathcal{D}$, any $\mu \in [a, b]$, and any $\delta \in (0, 1)$,*

$$\exists \rho \in \mathcal{P} : \mathbb{E}_\rho[\tilde{L}(h)] \geq \mu + (b - \mu)\mathrm{kl}^{-1,+}\left(\frac{\mathbb{E}_\rho[\hat{\tilde{L}}^+(h)]}{b - \mu}, \frac{\mathrm{KL}(\rho \,||\, \rho_0) + \ln(4\sqrt{N}/\delta)}{N}\right)$$
$$- (\mu - a)\mathrm{kl}^{-1,-}\left(\frac{\mathbb{E}_\rho[\hat{\tilde{L}}^-(h)]}{\mu - a}, \frac{\mathrm{KL}(\rho \,||\, \rho_0) + \ln(4\sqrt{N}/\delta)}{N}\right),$$

*with probability at most $\delta$.*

*Proof.* The theorem follows by applying Lemma A.2 to the decomposition $\mathbb{E}_\rho[\tilde{L}(h)] = \mu + \mathbb{E}_\rho[\tilde{L}^+(h)] - \mathbb{E}_\rho[\tilde{L}^-(h)]$. $\square$

### 2.2.2 Recursive PAC-Bayes bound

**Data-informed prior.** The tightness of PAC-Bayesian bounds is governed by the KL divergence between the posterior $\rho$ and the prior $\rho_0$. The better the prior guess, the tighter the bound. Because the prior must be independent of the observed data, a common choice is to select a prior as uniform as possible over the hypothesis space. To improve upon this naive choice, Ambroladze et al. (2006) proposed splitting the observed data into two disjoint subsets, $S_0$ and $S_1$, i.e., $\mathcal{D} = S_0 \cup S_1$, using $S_0$ to infer a *data-informed prior* and $S_1$ to subsequently evaluate the bound. This balances the benefit of a better prior with the cost of having fewer observations to evaluate the bound.

**Excess loss.** The *excess loss* $L^{\mathrm{exc}}(h)$ with respect to a reference hypothesis $h^* \in \mathcal{H}$ is defined as $L^{\mathrm{exc}}(h) = L(h) - L(h^*)$. The excess-loss concept allows the expected loss to be decomposed as $\mathbb{E}_\rho[L(h)] = \mathbb{E}_\rho[L(h) - L(h^*)] + L(h^*)$. Using $S_0$ to construct both the prior $\rho_0$ and the reference $h^*$, Mhammedi et al. (2019) showed that, assuming $L(h^*)$ is close to $L(h)$, the excess loss has lower variance and thus yields a more efficient bound, while a bound on $L(h^*)$ is independent of $\mathrm{KL}(\rho \,||\, \rho_0)$ and can be obtained using standard generalization guarantees.

**Recursive PAC-Bayes.** Wu et al. (2024) generalized the excess loss further by introducing a scaling factor $\kappa < 1$ to maintain a diminishing effect of recursions: $\mathbb{E}_\rho[L(h)] = \mathbb{E}_\rho[L(h) - \kappa\mathbb{E}_{\rho_0}[L(h^*)]] + \kappa\mathbb{E}_{\rho_0}[L(h^*)]$. Here, the first term reflects the excess loss with respect to a scaled version of the expected reference hypothesis loss under the prior $\rho_0$. The second term, in turn, is an expected loss similar to the one on the left-hand side of the equation. Instead of adhering to a binary split $\mathcal{D} = S_0 \cup S_1$ such that $S_0 \cap S_1 = \emptyset$, they proposed extending this decomposition recursively by partitioning $\mathcal{D}$ into $T$ disjoint subsets, $\mathcal{D} = \bigcup_{t=1}^{T} S_t$, and they define $S_{\leq t} = \bigcup_{s=1}^{t} S_s$ and $S_{\geq t} = \bigcup_{s=t}^{T} S_s$. Their recursion is given by

$$\mathbb{E}_{\rho_t}[L(h)] = \mathbb{E}_{\rho_t}\left[L(h) - \kappa_t\mathbb{E}_{\rho_{t-1}}[L(h)]\right] + \kappa_t\mathbb{E}_{\rho_{t-1}}[L(h)], \tag{2}$$

for $t \geq 2$, and $\kappa_1, \ldots, \kappa_T$ are scaling factors. The distributions $\rho_1, \ldots, \rho_T \in \mathcal{H}$ form a sequence such that $\rho_t$ depends solely on $S_{\leq t}$.

While Wu et al. (2024) formulated their final recursive bound directly for a zero-one loss and PAC-Bayes split-kl bounds (Wu & Seldin, 2022), we present their result first in a general, loss-agnostic form before constructing a specific bound in the next section.

**Theorem 2.2.** *(Recursive PAC-Bayes bound.) Let $\mathcal{D} = S_1 \cup \cdots \cup S_T$ be a disjoint decomposition of the set of observations $\mathcal{D}$. Let $S_{\leq t}$ and $S_{\geq t}$ be as defined above, $N = |\mathcal{D}|$, and $N_t = |S_{\geq t}|$. Let $\kappa_1, \ldots, \kappa_T$ be a sequence of scaling factors, where $\kappa_t$ is allowed to depend on $S_{\leq t-1}$. Let $\mathcal{P}_t$ be the set of distributions on $\mathcal{H}$ that are allowed to depend on $S_{\leq t}$, and $\rho_t \in \mathcal{P}_t$. Then, for any $\delta \in (0, 1)$,*

$$\mathbb{P}\left(\exists t \in [T], \rho_t \in \mathcal{P}_t \text{ such that } \mathbb{E}_{\rho_t}[L(h)] \geq \mathcal{B}_t(\rho_t)\right) \leq \delta,$$

*where $\mathcal{B}_t(\rho_t)$ is a generic PAC-Bayesian bound on $\mathbb{E}_{\rho_t}[L(h)]$ defined recursively as follows.*

$$\mathcal{B}_t(\rho_t) = \mathcal{E}_t(\rho_t, \kappa_t) + \kappa_t \mathcal{B}_{t-1}(\rho_{t-1}^*),$$

*where $\mathcal{B}_1(\rho_1)$ is a PAC-Bayes bound on $\mathbb{E}_{\rho_1}[L(h)]$ with an uninformed prior, and $\mathcal{E}_t(\rho_t, \kappa_t)$ is a PAC-Bayes bound on the excess loss $\mathbb{E}_{\rho_t}\left[L(h) - \kappa_t \mathbb{E}_{\rho_{t-1}^*}[L(h')]\right]$.*

*Proof.* Because $\mathcal{B}_1(\rho_1)$ and $\mathcal{E}_t(\rho_t, \kappa_t)$ are PAC-Bayes bounds by assumption, we have

$$\mathbb{P}\left(\exists \rho_1 \in \mathcal{P}_1 : \mathbb{E}_{\rho_1}[L(h)] \geq \mathcal{B}_1(\rho_1)\right) \leq \delta/T,$$
$$\text{and} \quad \mathbb{P}\left(\exists \rho_t \in \mathcal{P}_t : \mathbb{E}_{\rho_t}\left[L(h) - \kappa_t \mathbb{E}_{\rho_{t-1}^*}[L(h')]\right] \geq \mathcal{E}_t(\rho_t, \kappa_t)\right) \leq \delta/T \text{ for } t \in \{2, \ldots, T\}.$$

The claim follows by expected loss decomposition and the recursion. $\square$

## 3 Recursive PAC-Bayesian risk certificates for reinforcement learning

The general recursive PAC-Bayes framework introduced in Section 2.2.2 provides a generic method for certifying the expected loss of stochastic predictors. To apply this framework to reinforcement learning, hypotheses, data, and losses are interpreted in terms of policies, evaluation roll-outs, and return prediction. While we focus on actor-critic policies in our experiments, the certification pipeline described in steps (ii)–(iv) applies to any fixed stochastic policy $\pi$, regardless of how it was obtained.

Obtaining risk certificates involves four steps, following the conceptual structure in Figure 1.

***(i) Agent training.*** An actor–critic algorithm, e.g., REDQ (Chen et al., 2021), is trained until convergence or until a computational budget is exhausted. The resulting policy $\pi$ is then fixed. The objective is not to optimize $\pi$ further, but to certify its performance.

***(ii) Data collection.*** Execute the frozen policy for $N_{\text{ep}}$ episodes in evaluation mode (no exploration noise, no parameter updates). For each episode, store the trajectory $\tau$ and use it to compute discounted returns $G_t = \sum_{i=t}^{\infty} \gamma^{i-t} r_i$ for each visited state $s_t$. The discounted return is used as the prediction target rather than an undiscounted sum of rewards. Short-term risks tend to be more relevant for decisions, as longer-term risks depend on an increasing set of external, often unaccountable, factors. Discounted rewards also serve as a proxy for lifelong learning and policy evaluation, as they generalize to non-episodic data. Although the original policy might be trained on discounted returns in step *(i)*, a valid bound can also be constructed by computing undiscounted rewards from data collected in *(ii)*. As conservative samples are highly correlated, we apply a simple thinning strategy analogous to thinning in Markov chain Monte Carlo (MCMC). Subsampling of correlated samples in such chains reduces short-range autocorrelation without strictly enforcing independence. We sample by taking every $m$-th sample depending on the environment to better approximate the i.i.d. assumption required by PAC-Bayes and to form a data set $\mathcal{D} = \{(s_n, G_n)\}_{n=1}^N$. Our certificates should be interpreted as holding under this approximation. The experimental evaluation in Section 4 demonstrates empirically that, even under this approximation, the approach provides informative and well-correlated bounds.

As consecutive samples are highly correlated, a simple thinning strategy, described in the appendix, is applied to better approximate the i.i.d. assumption required by PAC-Bayes and to form a data set $\mathcal{D} = \{(s_n, G_n)\}_{n=1}^N$. Although a PAC-Bayesian bound tightens as the number of data points increases, it is observed that even a relatively small number of evaluation roll-outs is sufficient to obtain tight results.

***(iii) Fitting posteriors via recursive training.*** The prediction target is the discounted return $G_t$ observed at state $s_t$. The hypothesis $h$ is a Bayesian neural network (BNN) whose posterior $\rho_t$ is inferred by directly minimizing the PAC-Bayes bound $\mathcal{E}_t(\rho_t, \kappa_t)$ from Theorem 2.2, using a bounded loss $\ell$ normalized to $[0, B]$. This self-certified training ensures that the bound is tight by construction, guaranteeing that the predictor reliably estimates the policy's expected discounted return from any visited state and thereby providing a state-dependent risk certificate for the frozen policy $\pi$. To obtain data-informed priors that further tighten the bound, the data is partitioned into $T$ disjoint subsets $\mathcal{D} = S_1 \cup \cdots \cup S_T$ and a sequence of $T$ BNNs is trained, where the $t$-th network uses $\rho_{t-1}$ as a data-informed prior for $t > 1$ and an uninformative prior for $\rho_0$. We follow the experimental setup of Wu et al. (2024) and choose $\kappa_t = 0.5$ for all $t$ throughout our experiments. See Wu et al. (2024, Section 4) for a discussion on how to choose $\kappa_t$ via a grid-based approach.

***(iv) Risk certificate construction.*** Apply Theorem 2.2 to construct recursive bounds as follows.

**A bound for $\mathcal{B}_1$.** As $\hat{L}(h)$ is bounded in $[0, B]$, its expectation is rescaled, and

$$\mathcal{B}_1(\rho_1) = B\mathrm{kl}^{-1,+}\left(\frac{\mathbb{E}_\rho[\hat{L}(h)]}{B}, \frac{\mathrm{KL}\left(\rho_1 \parallel \rho_0^*\right) + \ln(2T\sqrt{N}/\delta)}{N}\right),$$

is chosen, where $\rho_0^*$ is a data-independent prior distribution on $\mathcal{H}$. Given the result in Theorem A.1, this is a PAC-Bayesian bound on $\mathbb{E}_{\rho_1}[L(h)]$, i.e.,

$$\mathbb{P}\left(\exists \rho_1 \in \mathcal{P}_1 : \mathbb{E}_{\rho_1}[L(h)] \geq \mathcal{B}_1(\rho_1)\right) \leq \delta/T.$$

**A bound for $\mathcal{E}_t$.** Let $L_t^{\mathrm{exc}}(h) = L(h) - \kappa_t\mathbb{E}_{\rho_{t-1}}[L(h')] \in [-\kappa_t B, B]$. For $\mu \in [-\kappa_t B, B]$, define $L_t^{\mathrm{exc}+}(h) = \max\{0, L_t^{\mathrm{exc}}(h) - \mu\}$ and $L_t^{\mathrm{exc}-}(h) = \max\{0, \mu - L_t^{\mathrm{exc}}(h)\}$, with $\hat{L}_t^{\mathrm{exc}+}(h)$ and $\hat{L}_t^{\mathrm{exc}-}(h)$ as their empirical analogues. Set

$$\mathcal{E}_t(\rho_t) = \mu + (B - \mu)\mathrm{kl}^{-1,+}\left(\mathbb{E}_{\rho_t}[\hat{L}_t^{\mathrm{exc}+}(h)]/(B - \mu), \Psi_t\right)$$
$$- (\mu + \kappa_t B)\mathrm{kl}^{-1,-}\left(\mathbb{E}_{\rho_t}[\hat{L}_t^{\mathrm{exc}-}(h)]/(\mu + \kappa_t B), \Psi_t\right),$$

where $\Psi_t = \mathrm{KL}\left(\rho_t \parallel \rho_{t-1}^*\right) + \ln(4T\sqrt{N_t}/\delta)/N_t$, and $\rho_{t-1}^*$ is a distribution on $\mathcal{H}$ informed by $S_{\leq t-1}$. Via Theorem 2.1, this is a PAC-Bayesian bound on $\mathbb{E}_{\rho_t}[L_t^{\mathrm{exc}}]$ that holds with probability at least $1 - \delta/T$, i.e.,

$$\mathbb{P}\left(\exists \rho_t \in \mathcal{P}_t \text{ such that } \mathbb{E}_{\rho_t}[L_t^{\mathrm{exc}}(h)] \geq \mathcal{E}_t(\rho_t)\right) \leq \delta/T.$$

Applying this construction recursively with $T$ steps therefore yields a recursive PAC-Bayesian bound that holds with probability at least $1 - \delta$.

**Why a learned value predictor?** The tightness of PAC-Bayes bounds is governed by the KL divergence between posterior and prior: the closer the prior is to the true solution, the tighter the bound. Achieving tight certificates therefore requires data-informed priors, i.e., priors that already approximate the value function well before the bound is evaluated. This, in turn, necessitates a parametric model whose prior can be shaped by data. We use a Bayesian neural network as this model: a portion of evaluation data is used to train an informed prior, and the remaining data is used to certify the posterior's generalization error via the PAC-Bayes bound. The bound is tight precisely because the data-informed prior mechanism keeps the KL divergence small.

## 4 EXPERIMENTS

The aim of our experiments is to address the following questions. *Q1*: Can the test-time return of a policy be predicted with high precision across a range of environments and policies with varying levels of expertise? *Q2*: What is the influence of the structure of a PAC-Bayesian bound? *Q3*: How does the validation set size affect the tightness of the risk certificate guarantee?

In all figures, *bound* refers to the PAC-Bayes upper bound on the expected squared return-prediction error of the BNN, and *test error* refers to the actual squared return-prediction error evaluated on held-out episodes not used during bound construction.

### 4.1 Experiment design

We evaluate our certificate-generation pipeline at an error tolerance of $\delta = 0.025$ on three popular representative actor-critic algorithms: PPO (Schulman et al., 2017), a classical and still widely used approach due to its robustness and simplicity despite comparatively low sample efficiency; SAC (Haarnoja et al., 2018a), a strong off-policy baseline that combines stability with high sample efficiency through entropy-regularized actor-critic learning; and REDQ (Chen et al., 2021), a more recent method that exemplifies advances in sample efficiency by leveraging randomized ensembles of Q-functions. Training and hyperparameter details for each method are provided in the appendix.

**Environments.** We rely on environments from three benchmarks. All three algorithms are evaluated in the main paper on three MuJoCo (Todorov et al., 2012) environments: *Half-Cheetah*, *Humanoid*, and *Hopper*, due to their widespread use in the community and the representative value of the platforms for real-world use cases. Additionally, REDQ is evaluated on three environments each from DM-Control (Tassa et al., 2018) another popular locomotion benchmark and MetaWorld (Yu et al., 2020), which focuses on robotic manipulation. The appendix includes REDQ results on two further MuJoCo environments.

**Data generation.** We train each agent for up to 300 000 steps, after which the policy is frozen. The learned policy is then run in evaluation mode for another 200 episodes, the first half of which are used to fit the bound, and the remainder serve as a test dataset to evaluate generalization performance. The discounted return on the test set is predicted by fitting a PAC-Bayes bound using the validation set.

**Policy instances.** We define a policy instance as the output of a single policy-training round. In our experiments, we consider five policy instances, each obtained by training with a different initial seed. Due to the stochastic nature of initialization and training, each instance follows a unique trajectory. We construct individual bounds for each instance and account for randomness in the risk-certificate generation process by repeating the procedure five times for every policy instance. To model policies of varying quality, we evaluate three training stages of each policy, each reflecting a different level of expertise: *Starter*, a policy trained for 100 000 steps; *Intermediate*, trained for 200 000 steps; and *Expert*, trained for 300 000 steps, after which no further performance improvements were observed.

**Baselines.** We design our baselines with the following goals: *(i)* how well a PAC-Bayes bound predicts test-time performance, *(ii)* whether informative priors yield tighter guarantees, *(iii)* whether the bound tightens when the recursive scheme is used, and *(iv)* whether increasing recursion depth improves tightness. As this is the first work to evaluate generalization bounds tailored to continuous control with deep actor-critics, there are no existing baselines for comparison. We consider two non-recursive baselines: *NonRec-NonInf*, a PAC-Bayes-kl inverse bound (see Theorem A.1) with a non-informative prior that is independent of the training data, and *NonRec-Inf*, a data-informed variant in which the dataset is split equally into $\mathcal{D} = \mathcal{D}_{\mathrm{prior}} \cup \mathcal{D}_{\mathrm{bound}}$, allowing the prior to depend on $\mathcal{D}_{\mathrm{prior}}$ and the empirical loss to be computed on $\mathcal{D}_{\mathrm{bound}}$. We evaluate our approach at two recursion depths, $T = 2$ (*Rec T=2*) and $T = 6$ (*Rec T=6*), to assess the effect of recursion.

**Performance metrics.** We evaluate the bounds using three metrics: *Normalized bound value*: To ensure comparability across environments with different reward scales, squared discounted return prediction errors are normalized by the maximum return observed during training. A value close to zero implies the bound closely follows the actual returns. *Tightness*: The difference between the predicted bound and the actual test error; smaller values indicate more accurate estimates of the discounted return prediction error. *Correlation*: A linear correlation is expected between the risk certificates and the observed test errors across policy instances.

We evaluate and compare the final posterior loss $\rho$ on full training data and held-out test data, alongside the corresponding PAC-Bayes bounds across all methods and environments. To mitigate overfitting common in continuous-control settings, where consecutive samples are highly correlated, we apply a thinning strategy that reduces redundancy while preserving data diversity. Full details for each experiment are provided in Section D. We provide an implementation at `anonymous`.

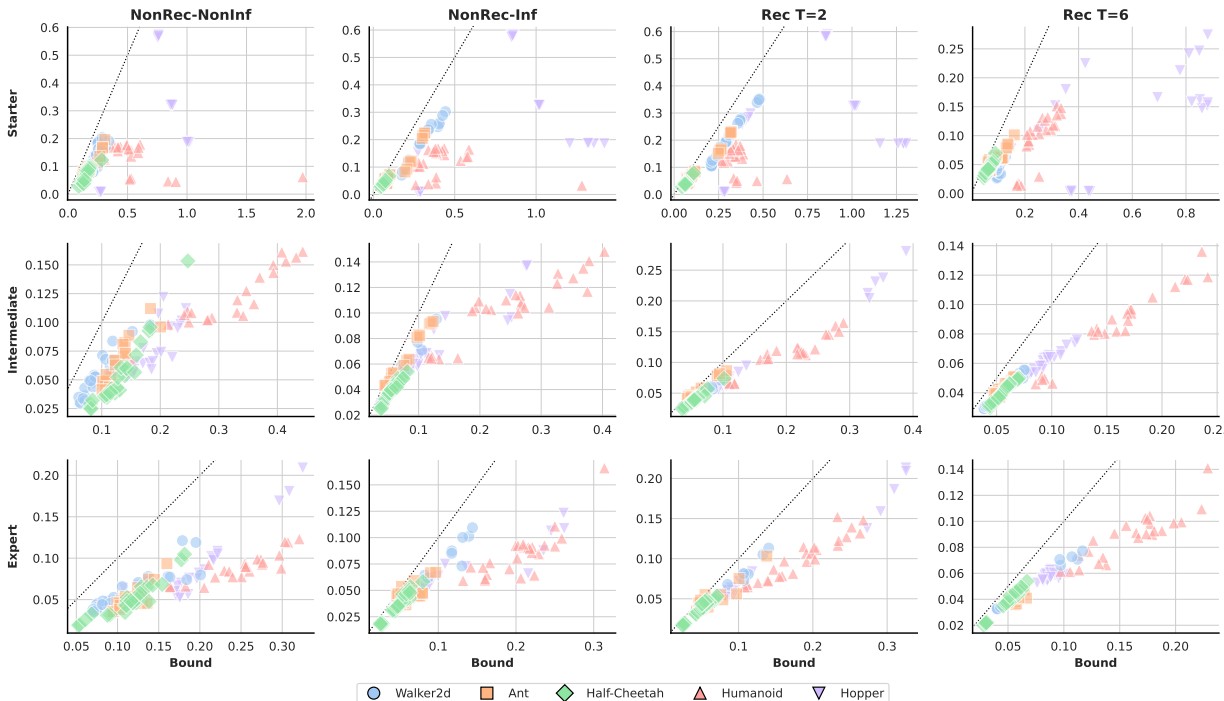

Figure 2: *Correlation between bounds and test errors.* PAC-Bayes bounds (x-axis) are plotted axis against true test errors (y-axis) for REDQ across five MuJoCo environments, policy instances, and repetitions to visualize correlation. We observe a high correlation, especially as policies improve and bounds become recursive.

## 4.2 Results

Comprehensive results for every environment, policy instance, and repetition are presented in Section E. The main text focuses on aggregated results.

**Q1: Can the test-time return of a policy be predicted with high precision?** Figures 2 to 4 presents scatter plots of all PAC-Bayesian bounds discussed in 4.1, alongside policy instances and repetitions, against their respective test set errors across environments and levels of policy expertise for REDQ. On MuJoCo, for every bound, the correlation between the bound and the test error increases with policy expertise. Within a fixed expertise level, the correlation also improves as the bound becomes more advanced, a trend already evident for noisier starter policies. For example, in the brittle Hopper environment, which exhibits the weakest correlations overall, moving from NonRec-NonInf to Rec with $T = 6$ raises the Pearson correlation from 0.4 to 0.65. At higher expertise levels, the recursive bounds achieve correlations above 0.9 in almost all environments. MetaWorld shows the same overall trend for two out of three environments, with Reacher as an outlier that shows a correlation pattern between bound and test error, but less clear. Finally, DM-Control shows less correlation than the non-recursive baselines, as the recursion increases for the Window open environment, however at a greater overall tightness.

Overall, there is a strong correlation between bounds and test errors. There is also increasing scatter as the expertise level decreases. This is expected, as the effects of an unconverged policy on environment dynamics are less predictable. The bounds therefore provide a reliable prediction of the test-time return, thereby answering Q1. See the appendix for the corresponding correlation plots for the other agent and benchmark combinations.

> **Answer:** *Bounds and test errors are strongly correlated.*

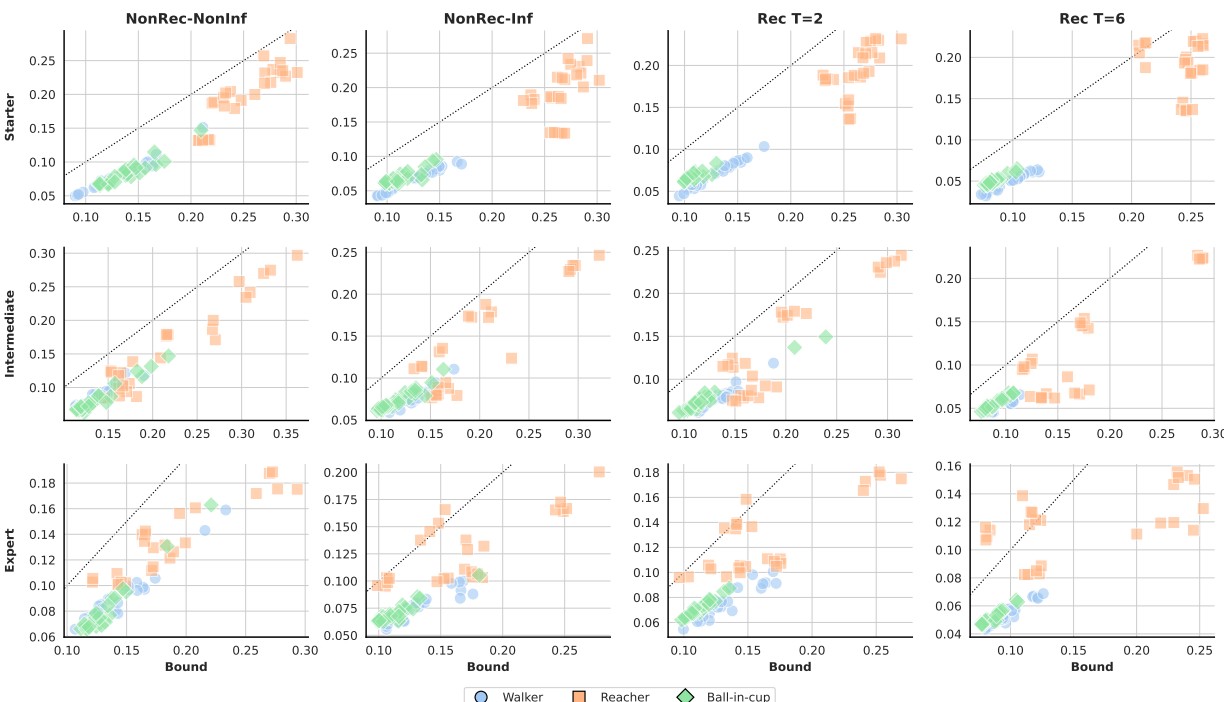

Figure 3: *Correlation between bounds and test errors.* PAC-Bayes bounds (x-axis) are plotted axis against true test errors (y-axis) for REDQ across three MetaWorld environments, policy instances, and repetitions to visualize correlation. We observe a high correlation, especially as policies improve and bounds become recursive.

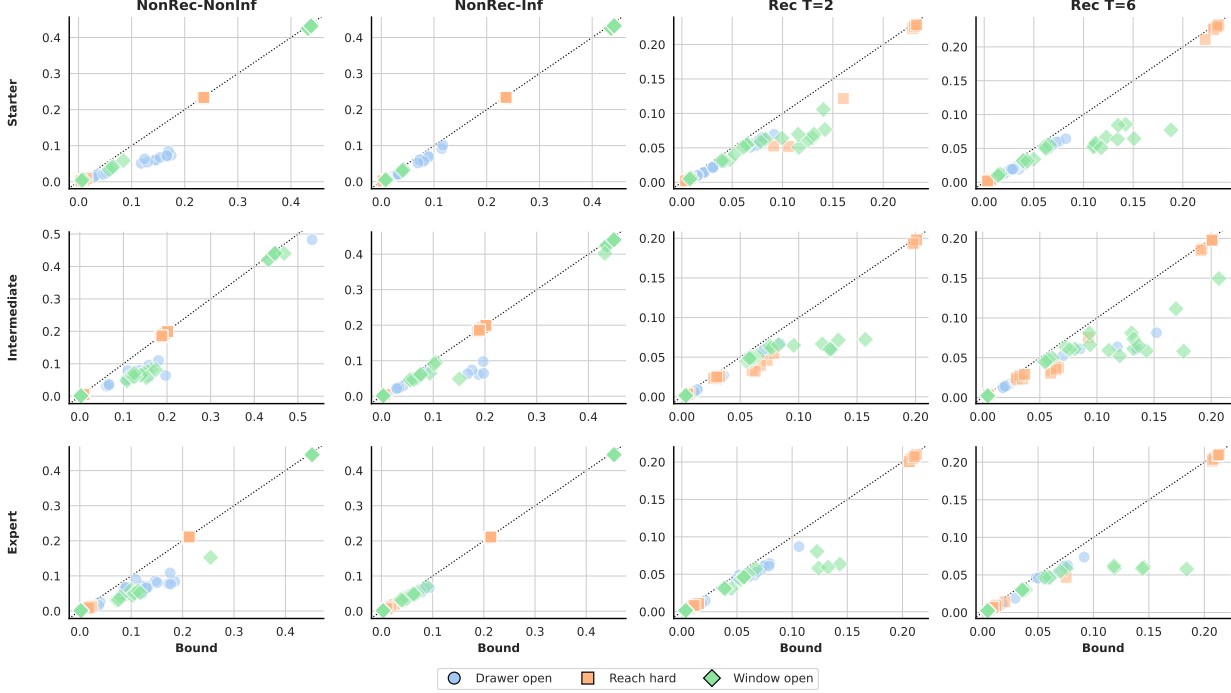

Figure 4: *Correlation between bounds and test errors.* PAC-Bayes bounds (x-axis) are plotted axis against true test errors (y-axis) for REDQ across three DM-Control environments, policy instances, and repetitions to visualize correlation. We observe a high correlation, especially as policies improve and bounds become recursive.

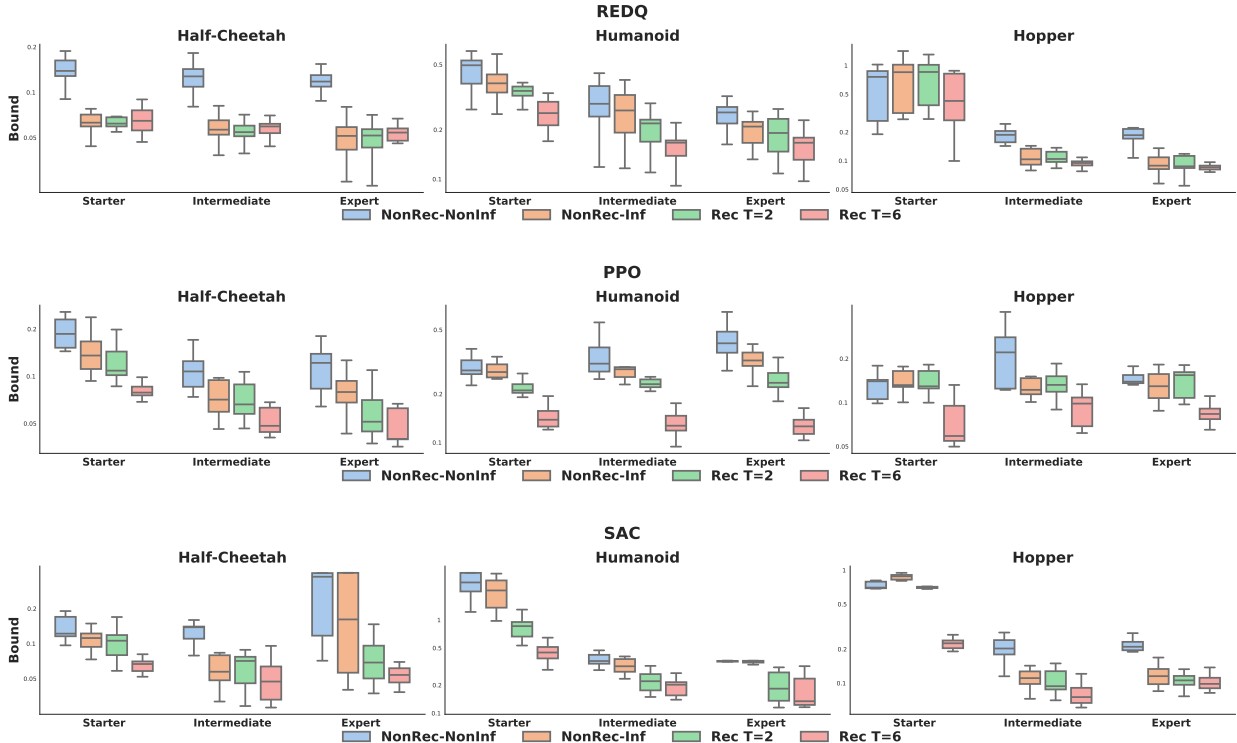

Figure 5: *Bound values.* Normalized bound values for all baselines across three MuJoCo environments and all policy quality levels. Results are aggregated across all seeds and repetitions.

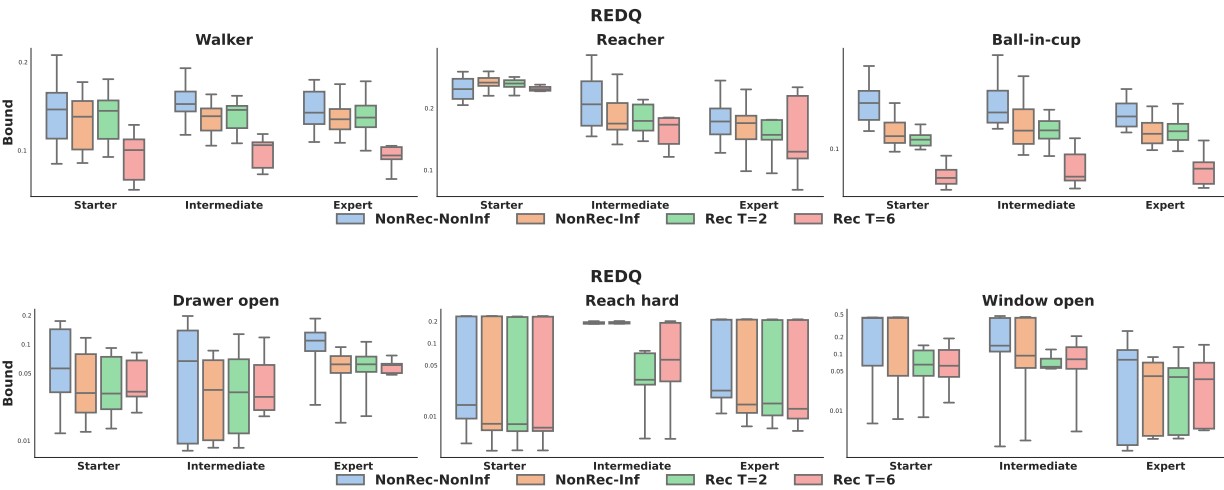

Figure 6: *Bound values.* Normalized bound values for REDQ across DM-Control *(upper plot)* MetaWorld *(lower plot)* and all policy quality levels. Results are aggregated across all seeds and repetitions.

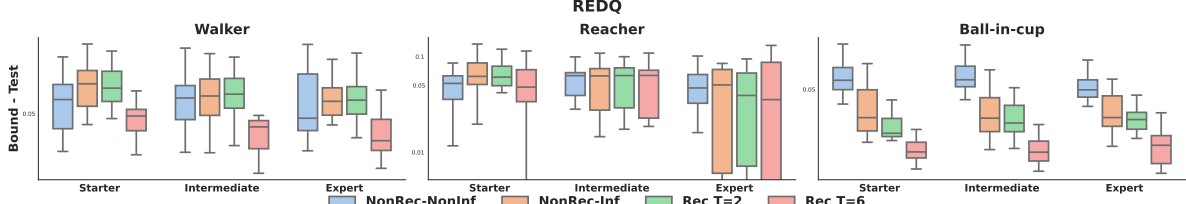

Figure 7: *(Bound - test) values.* Tightness of the bound over all baselines and policy levels considered for REDQ across five MuJoCo environments. The plots aggregated over policy instances and their repetitions.

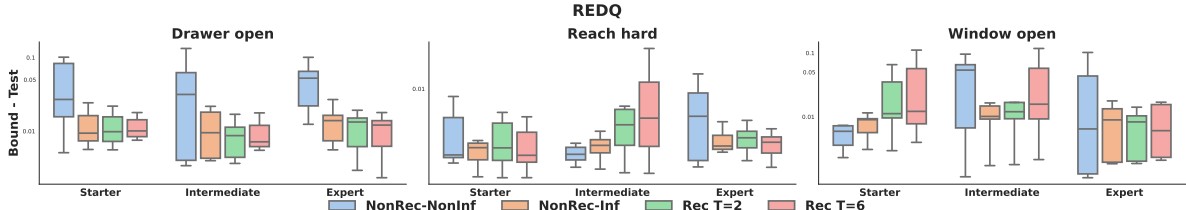

Figure 8: *(Bound - test) values.* Tightness of the bound over all baselines and policy levels considered for REDQ across three DM Control environments. The plots aggregated over policy instances and their repetitions.

**Q2: What is the influence of a PAC-Bayes bound's structure?** In Figure 5 we plot the normalized bounds aggregated over policy instances and repetitions for each of the three models on three MuJoCo environments. Data-informed priors usually improve bounds across environments for all policy levels and actor-critic methods. Introducing recursion further tightens bounds, with deeper recursion generally yielding the tightest results. Figure 6 shows the same pattern for REDQ on DM-Control, while the bounds on MetaWorld already converge after the initial introduction of prior knowledge, i.e., $T = 1$. We assume this to be due to the inherent complexity and inter-seed variability of MetaWorld environments, with Reacher, as in Figure 4, being a clear outlier.

Figures 7 to 11 show the tightness of all baselines, measured as the difference between the normalized bound score and the normalized test error. Values are aggregated across policy instances and repetitions for each of the five environments, with smaller values indicating tighter bounds. Box plots display the distribution of these differences: the box covers the interquartile range (25th to 75th percentile), the median is marked inside, and whiskers extend to points within 1.5 times the interquartile range, illustrating typical variability. Outliers beyond this range are omitted for clarity. This effectively captures both the central tendency and the dispersion of the data across policies and methods. The plot indicates that data-informed priors generally yield tighter bounds, although this effect is less pronounced at the Starter level. Moreover, applying recursion and increasing its depth further tighten the bounds. The improvements due to recursion are more noticeable in challenging environments and less so in simpler tasks.

Figure 9: *(Bound - test) values.* Tightness of the bound over all baselines and policy levels considered for REDQ across three Meta-world environments. The plots aggregated over policy instances and their repetitions.

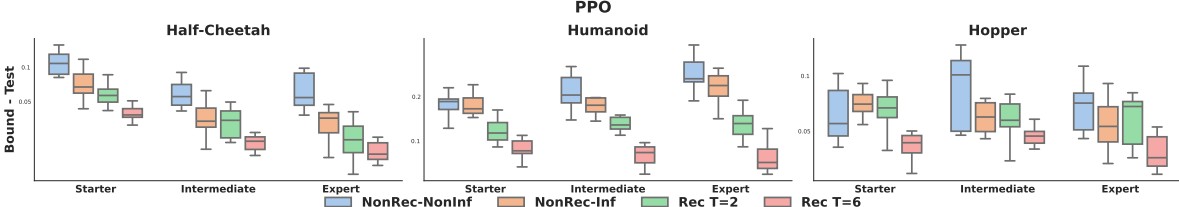

Figure 10: *(Bound - test) values.* Tightness of the bound over all baselines and policy levels considered for PPO across three MuJoCo environments. The plots aggregated over policy instances and their repetitions.

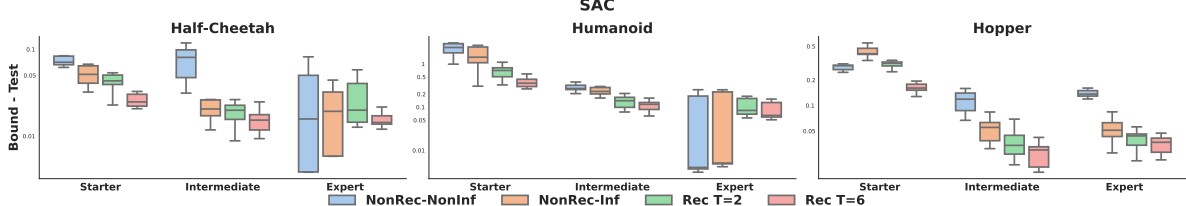

Figure 11: *(Bound - test) values.* Tightness of the bound over all baselines and policy levels considered for SAC across three MuJoCo environments. The plots aggregated over policy instances and their repetitions.

**Answer:** *Bound tightness improves with increasing recursive depth.*

**Q3: How does the validation set size influence the tightness of the risk certificate guarantee?** Collecting validation data from physical robots is often costly. Hence, the sample efficiency of a risk-certificate generation pipeline is of particular interest. Figure 12 presents the tightness scores of the bounds across varying validation set sizes for REDQ in the Humanoid environment, while the test set size is kept fixed. As expected, larger validation sets yield tighter bounds, with the effect most pronounced for the proposed recursive bounds. A recursion with depth $T = 6$ achieves tightness comparable to that of the nonrecursive bounds while requiring only half as many data points. These findings demonstrate that recursive bounds substantially improve sample efficiency, addressing Q3.

**Answer:** *Recursion improves sample efficiency.*

**Local reparameterization improves tightness.** To train our model, we use a Bayesian neural network (BNN) that represents uncertainty by learning distributions over parameters. To our knowledge, prior work on PAC-Bayesian risk certification with BNNs has relied exclusively on Blundell et al. (2015)'s *Bayes by backprop* approach (see, e.g., Pérez-Ortiz et al., 2021). We show in Figure 13 that using the *local reparameterization trick (LRT)* (Kingma et al., 2015) to compute the empirical risk term in the bound calculation substantially improves the tightness of all four evaluated bounds. This effect holds even for the already saturated expert-level policy in the challenging Humanoid environment. Further details can be found in Section E.

## 5 Limitations, future work, and broader impact

We restricted our empirical investigation to three actor-critic algorithms on three benchmark suites. This was a deliberate choice to facilitate interpretation and maintain feasibility. We do not expect meaningful additional information from extending the same pipeline to additional RL agents and benchmarks. The next major step would be to implement our pipeline on a physical platform under controlled conditions. The applicability of our findings to more advanced control settings, such as sparse-reward scenarios that require goal-conditioned or hierarchical RL algorithm design, is subject to further investigation. We leave

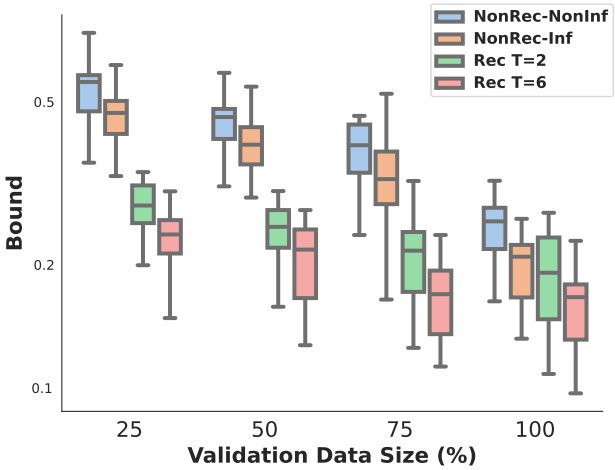

Figure 12: *Effect of validation data size on tightness.* Bounds for REDQ on Humanoid with an expert policy.

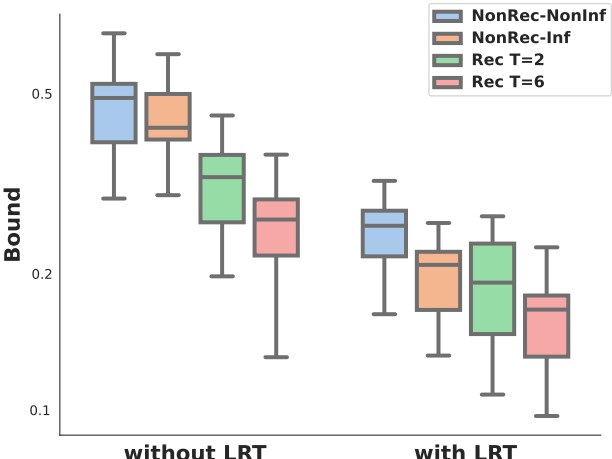

Figure 13: *Local reparameterization trick (LRT).* Influence for REDQ on Humanoid with an expert policy.

this enterprise to future work, as deep learning-based solutions for such setups have not yet reached sufficient maturity to move beyond simulations. Another significant advance would be to proceed from our current self-certified policy evaluation approach to self-certified policy optimization in an online setting. This would necessitate training the policy via a PAC-Bayes bound. However, RL is a feedback-loop system in which ensuring convergence, numerical stability, and optimal trade-offs between exploration and exploitation are major determinants of stable training. While promising preliminary results exist (Tasdighi et al., 2024; 2025), the problem is fundamental and requires a dedicated research program—an effort that goes beyond the scope of a single paper.

Our work contributes to the trustworthy development of agentic AI technologies, thereby promoting their adoption by society. Public concerns about such technologies will be even more pronounced when they are deployed on physical systems in direct contact with humans. Thanks to reliable risk certificates, such safety-critical technologies are likely to receive wider adoption. This, in turn, will further accelerate their development by expanding the pool of practice and observations.

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

# A  Theory

## A.1  PAC-Bayes-kl bound

Assuming the definitions in Section 2.2, we cite the following bound and kl-inequality. See the respective references for further details.

**Theorem A.1** (PAC-Bayes-kl bound (Seeger, 2002; Maurer, 2004)). *For any probability distribution $\rho_0 \in \mathcal{P}$ that is independent of $\mathcal{D}$ and any $\delta \in (0,1)$, we have*

$$\mathbb{P}\left(\exists \rho \in \mathcal{P} : \mathrm{kl}\big(\mathbb{E}_\rho[\hat{L}(h)]\|\mathbb{E}_\rho[L(h)]\big) \geq \big(\mathrm{KL}\left(\rho \| \rho_0\right) + \ln(2\sqrt{N}/\delta)\big)/N\right) \leq \delta.$$

*Proof.* See, e.g., Maurer (2004) for a proof of the bound. □

**Lemma A.1** (kl-inequality (Langford, 2005; Foong et al., 2021; 2022)). *Let $Z_1, \ldots, Z_N$ be i.i.d. random variables taking values in an interval $[0,1]$, and let $\mathbb{E}[Z_n] = p$ for all $n$. Let their empirical mean be $\hat{p} = \frac{1}{N}\sum_{n=1}^N Z_n$. Then, for any $\delta \in (0,1)$, we have*

$$\mathbb{P}\left(\mathrm{kl}(\hat{p}\|p) \geq \ln(1/\delta)/N\right) \leq \delta,$$

*the inverse of which is given by*

$$\mathbb{P}\left(p \geq \mathrm{kl}^{-1,+}\left(\hat{p}, \ln(1/\delta)/N\right)\right) \leq \delta,$$

*and*

$$\mathbb{P}\left(p \leq \mathrm{kl}^{-1,-}\left(\hat{p}, \ln(1/\delta)/N\right)\right) \leq \delta.$$

*Proof.* See Langford (2005), Corollary 3.7, for a proof of the bound. □

## A.2  Split-kl inequality

Let $Z \in [a,b]$, with $a,b \in \mathbb{R}$, be a random variable, and let $p = \mathbb{E}[Z]$. For $\mu \in [a,b]$, define $Z^+ = \max\{0, Z - \mu\}$ and $Z^- = \max\{0, \mu - Z\}$ so that $Z = \mu + Z^+ - Z^-$. Let $p^+ = \mathbb{E}[Z^+]$ and $p^- = \mathbb{E}[Z^-]$ be their respective expectations, and let $\hat{p}^+ = \frac{1}{N}\sum_{n=1}^N Z_n^+$ and $\hat{p}^- = \frac{1}{N}\sum_{n=1}^N Z_n^-$ be their empirical means for an i.i.d. sample $Z_1, \ldots, Z_N$.

**Lemma A.2** (Split-kl inequality (Wu & Seldin, 2022)). *For any $\mu \in [a,b]$ and $\delta \in (0,1)$,*

$$\mathbb{P}\left(p \leq \mu + (b-\mu)\mathrm{kl}^{-1,+}\left(\frac{\hat{p}^+}{b-\mu}, \frac{\ln(2/\delta)}{N}\right) - (\mu-a)\mathrm{kl}^{-1,-}\left(\frac{\hat{p}^-}{\mu-a}, \frac{\ln(2/\delta)}{N}\right)\right) \geq 1 - \delta.$$

*Proof.* The lemma follows by applying Lemma A.1 to each of the kl terms and a union bound. □

# B  Related work

**Theoretical analysis of reinforcement learning.** Exploration in finite-horizon Markov decision processes (MDPs) has been addressed in several prior works. UCBVI (Azar et al., 2017) offers a simple approach based on optimism under uncertainty, combining computational efficiency with conceptual clarity for exploration in finite-horizon MDPs. BayesUCBVI (Tiapkin et al., 2022b) builds on this by introducing a tabular, stage-dependent, episodic reinforcement learning algorithm that uses a quantile of the posterior distribution of Q-values for optimism, eliminating the need for explicit bonus terms. However, it remains uncertain whether the PSRL (Osband et al., 2013) approach can achieve optimal problem-independent regret bounds. UCRL2 (Auer et al., 2008), an extension of UCRL (Auer & Ortner, 2006), employs upper confidence bounds to guide policy selection, encouraging exploration of uncertain state–action pairs and reducing regret over time, resulting in a near-optimal regret bound that grows logarithmically with the number of episodes. Agrawal & Jia (2017) propose a PSRL-inspired algorithm for multi-armed bandits, using posterior sampling

to incorporate uncertainty and to form an optimistic policy, achieving near-optimal regret bounds. It remains unclear whether it can achieve the problem-independent lower bound. OPSRL (Tiapkin et al., 2022a) extends PSRL by using multiple posterior samples instead of a single one, providing high-probability regret-bound guarantees and ensuring strong performance.

**PAC-Bayes analysis.** PAC-Bayesian analysis provides a frequentist framework for integrating prior knowledge into learning algorithms (Shawe-Taylor & Williamson, 1997; McAllester, 1998; Alquier et al., 2024). It leverages priors that sustain high confidence during learning (McAllester, 1999; Seeger, 2002; Catoni, 2007; Thiemann et al., 2017; Rivasplata et al., 2019; Pérez-Ortiz et al., 2021), but the resulting confidence intervals rely on data excluded from prior formation, making prior data allocation a critical challenge. Strategies for crafting effective PAC-Bayesian priors address this limitation (Ambroladze et al., 2006; Dziugaite et al., 2021; Wu et al., 2024).

**PAC-Bayesian risk certificate.** In recent years, PAC-Bayesian methods have also been used to compress neural nets (Lotfi et al., 2022) and to provide risk certificates for both uninformed (Dziugaite & Roy, 2017) and data-informed priors (Dziugaite & Roy, 2018; Dziugaite et al., 2021; Pérez-Ortiz et al., 2021). Reeb et al. (2018) introduce PAC-Bayesian bounds for Gaussian process-based regression, and an increasing body of literature now provides risk certificates for deep generative models, such as VAEs (Mbacke et al., 2023b), GANs (Mbacke et al., 2023a), and contrastive learning frameworks (Nozawa et al., 2020). Hsu et al. (2022) use PAC-Bayes to provide realistic safety guarantees, focusing primarily on worst-case risk and out-of-distribution safety.

**PAC-Bayesian RL.** PAC analysis in RL has led to algorithms that provide formal guarantees on sample complexity and performance. Strehl et al. (2006) introduced Delayed Q-Learning, an efficient model-free RL method. Strehl et al. (2009) established PAC bounds for both model-free and model-based methods in finite MDPs, highlighting sample efficiency. For complex observation spaces, Krishnamurthy et al. (2016) proposed Least-Squares Value Elimination, extending contextual bandit frameworks to sequential settings with PAC guarantees. In model-based RL, Dann et al. (2017) introduced the Uniform-PAC framework for finite-state episodic MDPs, achieving optimal sample complexity and regret bounds. Jiang (2018) improved model-based efficiency with a polynomial-complexity algorithm. PAC-Bayesian frameworks have also been applied in RL for policy evaluation and stability. Fard et al. (2011) introduced a PAC-Bayesian method for policy evaluation with probabilistic guarantees, which was subsequently extended to soft actor–critic approaches (Tasdighi et al., 2024) and deep exploration with sparse rewards (Tasdighi et al., 2025). These methods are designed exclusively for policy optimization, using PAC-Bayesian principles as algorithmic guidelines rather than for reliable risk certificate generation, which is our objective. Both methods employ approximately calculated PAC-Bayesian bounds without quantifying the resulting approximation errors. While such approximations are acceptable in policy search contexts, where learning performance is the main concern, they preclude rigorous analytical statements about test-time performance and therefore cannot support risk certification. Despite these advancements, the tightness, that is, the quality, of risk certificates remained an open question.

**Off-policy policy evaluation.** Off-policy policy evaluation (OPE) methods (Precup et al., 2000; Thomas & Brunskill, 2016; Jiang & Li, 2016; Dudík et al., 2014) estimate the value of a target policy from data collected under a different behavior policy, typically using importance sampling. Our setting differs in that we collect fresh on-policy rollouts from the frozen policy, which avoids the high variance of importance-weighted estimators. OPE is advantageous when additional data collection is infeasible, whereas our approach is designed for settings where a modest number of evaluation episodes can be collected, a realistic assumption for the robotic deployment scenarios we target. The two approaches are complementary: OPE methods could be used to reuse training data, while our PAC-Bayesian certificates provide formally guaranteed bounds on a learned value predictor.

## C Algorithms

In this section, two pseudocodes are presented. Algorithm 1 illustrates the overall pipeline of the method, capturing the main components from policy training to PAC-Bayes bound calculation.

Algorithm 2 presents the Recursive PAC-Bayes framework in detail, outlining the step-by-step calculation of the recursive bound across different splits of the training data, ultimately yielding the final bound. This framework supports multiple levels of recursion (e.g., two or six levels in our experiments). Details on the size of each data split and the selection strategy are provided in Section D.

---

**Algorithm 1** Predicting Policy Return via PAC-Bayes Bound Fitting

---

1: **Input:** Environment, actor-critic algorithm, BNN architecture, PAC-Bayes bounds
2: **Output:** Predicted test-time discounted returns and PAC-Bayes generalization bounds

    **Policy Training**
3: Train policy $\pi$ using the REDQ actor-critic algorithm.
4: Save policy parameters after specified training steps.

    **Data Collection**
5: Disable further learning.
6: Execute the saved policy for a few roll-outs and collect data.
7: Split the data into training and test sets.

    **Bayesian Model Training**
8: Initialize a Bayesian neural network (BNN).
9: Train the Bayesian model on the training data using a PAC-Bayes-inspired loss as the training objective.

    **Bound Construction and Evaluation**
10: **for** each bound type **do**
11:     Specify prior and posterior distributions.
12:     Compute the PAC-Bayes bound using model-predicted outputs.
13: **end for**

14: **Return:** Model predictions on the test data and bound evaluations on the training data.

---

# D Experimental details

In this section, we present the details and design choices for our environments, dataset, model training, and the replication of our experimental results. We consider five environments from version 'v4' of the MuJoCo suite (Todorov et al., 2012) (ant, half-cheetah, hopper, walker2d, humanoid), three environments from DM Control (Tassa et al., 2018) (ball-in-cup, reacher, walker), and three from Meta-World (Yu et al., 2020) (drawer-open, window-open, reach). All implementations in this work utilize the PyTorch framework (Paszke et al., 2019), version '2.5.1', and the OpenAI Gym environment (Brockman, 2016), version '0.29.0'.

## D.1 Specific hyperparameters

For recursive PAC-Bayes, we use $\kappa_t = 1/2$ for all $t$ and study the impact of recursion for data split over different portions. In our experiments, we set $\delta = 0.025$, following Wu et al. (2024)'s implementation.[3]

In addition to the primary confidence parameter $\delta$, which controls the overall probability with which the PAC-Bayes bound holds, we introduce an auxiliary confidence parameter $\delta'$, following Wu et al. (2024). This parameter accounts for the approximation error introduced when estimating the expected empirical loss and empirical excess loss via sampling (Pérez-Ortiz et al., 2021), since these quantities lack closed-form expressions in our setting. Specifically, $\delta'$ ensures that these sample-based estimates are accurate with high probability. We set $\delta' = 0.01$, consistent with the choice in the referenced work, and apply a union bound to combine the confidence levels. As a result, the overall bound holds with probability at least $1 - \delta - \delta'$, covering both the generalization guarantee and the estimation accuracy. This consideration is applied only during the evaluation phase for estimating the bounds, not during the optimization process.

---

[3]https://github.com/pyijiezhang/rpb

---

**Algorithm 2** Recursive PAC-Bayes bound loss computation

---

1: **Input:** Training dataset split $\mathcal{D} = S_1 \cup \cdots \cup S_T$ with total size $N = |\mathcal{D}|$, where $N_t = |S_{\geq t}|$; scaling factors $\kappa_1, \ldots, \kappa_T$; posterior parameters $\{\rho_1, \ldots, \rho_T\}$; and fixed confidence levels $\delta$, $\delta'$.

2: Calculate the loss for the first portion: $\hat{L}(h) \in [0, B]$

$$\mathbb{E}_{\rho_1}[\hat{L}^+(h)] \leq \mathrm{kl}^{-1,+}\left(\frac{1}{N_t}\sum\left(\max\left\{0, \hat{L}(h) - \mu\right\}\right), \frac{\ln(T/\delta')}{N_t}\right).$$

3: Specify the first portion of the recursive bound:

$$\mathcal{B}_1(\rho_1) = B\mathrm{kl}^{-1,+}\left(\frac{\mathbb{E}_{\rho_1}[\hat{L}^+(h)]}{B}, \frac{\mathrm{KL}\left(\rho_1 \parallel \rho_0^*\right) + \ln(2T\sqrt{N}/\delta)}{N}\right).$$

4: **for** each portion from $t = 2$ to the end **do**

5:  Calculate the excess loss:

$$L_t^{\mathrm{exc}}(h) = L(h) - \kappa_t\mathbb{E}_{\rho_{t-1}}\left[L(h')\right] \in [-\kappa_t B, B].$$

6:  Then,

$$\mathbb{E}_{\rho_t}[\hat{L}_t^{\mathrm{exc}+}(h)] \leq \mathrm{kl}^{-1,+}\left(\frac{1}{N_t}\sum\left(\max\left\{0, \hat{L}_t^{\mathrm{exc}}(h) - \mu\right\}\right), \frac{\ln(2T/\delta')}{N_t}\right),$$

$$\mathbb{E}_{\rho_t}[\hat{L}_t^{\mathrm{exc}-}(h)] \leq \mathrm{kl}^{-1,-}\left(\frac{1}{N_t}\sum\left(\max\left\{0, \mu - \hat{L}_t^{\mathrm{exc}}(h)\right\}\right), \frac{\ln(2T/\delta')}{N_t}\right).$$

7:  Compute $\mathcal{E}_t(\rho_t)$:

$$\mathcal{E}_t(\rho_t) = \mu + (B - \mu)\mathrm{kl}^{-1,+}\left(\frac{\mathbb{E}_{\rho_t}[\hat{L}_t^{\mathrm{exc}+}(h)]}{B - \mu}, \frac{\mathrm{KL}\left(\rho_t \parallel \rho_{t-1}^*\right) + \ln(4T\sqrt{N_t}/\delta)}{N_t}\right)$$

$$- (\mu + \kappa_t B)\mathrm{kl}^{-1,-}\left(\frac{\mathbb{E}_{\rho_t}[\hat{L}_t^{\mathrm{exc}-}(h)]}{\mu + \kappa_t B}, \frac{\mathrm{KL}\left(\rho_t \parallel \rho_{t-1}^*\right) + \ln(4T\sqrt{N_t}/\delta)}{N_t}\right).$$

8:  Update the bound iteratively:

$$\mathcal{B}_t(\rho_t) = \mathcal{E}_t(\rho_t, \kappa_t) + \kappa_t\mathcal{B}_{t-1}(\rho_{t-1}^*).$$

9:  $t = t + 1$

10: **end for**

11: With probability at least $1 - \delta$,

$$\mathbb{E}_{\rho_t}\left[L(h)\right] \leq \mathcal{B}_t(\rho_t).$$

---

## D.2 Constructing validation and test datasets

Our methodology comprises two phases: *training* and *evaluation*. To collect the transition data required for constructing validation and test datasets, we train an actor-critic agent (SAC, PPO, or REDQ).

Training is conducted for 300 000 environment steps, after which the resulting policy is saved as the *expert* policy. We also snapshot the policy at 100 000 and 200 000 steps, referring to them as the *starter* and *intermediate* policies, respectively, to enable analysis across varying levels of agent proficiency.

In the evaluation phase, learning is disabled, and the agent is executed with exploration turned off and no policy updates performed. Using the frozen policy (at each of the three proficiency levels), we collect 100 episodes of state transitions and rewards to construct a validation dataset for fitting the PAC-Bayes bound. An additional 100 episodes are collected under the same conditions to form the test dataset. This test data is used to compute a proxy for generalization performance by evaluating the predicted discounted returns of the frozen policy on previously unseen trajectories.

## D.3 Neural network architectures

**REDQ agent training.** We use an ensemble of ten critic networks and a single actor network. Each neural network consists of three layers, with layer normalization after each layer to regularize the network, following the approach of Ball et al. (2023). Additionally, we use the concatenated ReLU activation function (Shang et al., 2016), which combines the positive and negative parts of two ReLU activations and concatenates them. This leads to richer feature representations and enables the learning of more complex patterns.

During training, we employ 10 000 warm-up steps without policy updates, during which the agent only interacts with the environment; during the evaluation phase, we do not use any warm-up steps. We use a replay ratio (RR) of one in both the training and evaluation phases, which provides training stability and sample efficiency. We employ the Huber loss in the critic to measure the discrepancy between predicted Q-values and target Q-values, as it consistently provides performance advantages across various baselines. Furthermore, the temperature parameter $\alpha$ is automatically tuned during training to regulate policy entropy and balance exploration and exploitation, following the method introduced by Haarnoja et al. (2018b).

All these architectural and training choices were empirically found to improve the model's overall performance. Table 1 summarizes the hyperparameters and network configurations used in our experiments.

**PPO agent training.** Proximal Policy Optimization is an on-policy algorithm that trains a single actor network together with a single critic network representing the state-value function. The policy is updated using a clipped surrogate objective to prevent overly large updates, which improves stability but makes PPO less sample-efficient than off-policy methods. In our experiments, we followed the standard PPO baseline and evaluated performance on three widely used MuJoCo environments of varying difficulty and dimensionality: `HalfCheetah, Humanoid`, and `Hopper`. Since PPO typically requires more training to converge, we extended the training horizon to 1,000,000; 3,000,000; and 5,000,000 steps to obtain starter, intermediate, and expert policies, respectively. Risk bounds were computed in the same manner as in the REDQ experiments.

**SAC agent training.** Soft Actor-Critic is an off-policy algorithm that combines a single actor network with two Q-function critic networks, following the standard practice to mitigate overestimation bias. SAC is generally more sample-efficient than on-policy methods and stabilizes learning through target networks and soft updates. We implemented SAC in its canonical form and considered three training levels: 50,000 steps (starter), 100,000 steps (intermediate), and 1,000,000 steps (expert), using the same environments as above. The corresponding policies were then evaluated under our PAC-Bayesian risk bounds, consistent with the REDQ setting.

For both methods, the actor and critic networks follow the same neural network architecture described in Section D.3 for REDQ.

**Bayesian model training.** We aim to predict the test-time discounted return of a policy $\pi$ by fitting a PAC-Bayes generalization bound using training data. To this end, we use a Bayesian neural network (BNN), which allows us to capture model uncertainty and compute valid generalization guarantees.

The BNN is a feedforward neural network with two hidden layers and one output layer. Each layer is implemented as a variational Bayesian linear layer, where weights are modeled as independent Gaussian distributions with learnable means and variances. ReLU activations are used between layers to introduce nonlinearity.

During training, we apply the local reparameterization trick (Kingma et al., 2015), which samples the layer outputs rather than the weights. This reduces the variance of the gradient estimates, improves training stability, and reduces computational cost.

We train the BNN using a PAC-Bayes-inspired objective based on McAllester's bound (McAllester, 1998), which balances empirical loss and a KL divergence term. The KL term acts as a regularizer that penalizes deviation from a prior distribution, effectively controlling model complexity. This enables us to compute reliable generalization bounds and estimate the expected test-time return of the policy. Architectural details are summarized in Table 1.

### D.4  Further details on the bounds

We evaluate two non-recursive baselines and two recursive variants with different depths.

For the uninformed baseline (NonRec-NonInf), we use a PAC-Bayes-kl inverse bound in which the prior is initialized without dependence on the training data. The posterior is learned using the entire training dataset, and the bound is evaluated on this full dataset. For the informed case (NonRec-Inf), we split the training dataset in half: the first half is used to learn a data-informed prior, and the second to learn the posterior, with both steps minimizing a PAC-Bayes bound. The bound is evaluated using the same subset used to learn the posterior.

In the recursive settings, we apply these splits iteratively. For example, with recursion depth two, we divide the data into two 50–50 splits. For deeper recursion with six steps, we split the training set into increasingly smaller partitions based on the number of episodes $[3, 4, 6, 12, 25, 50]$. This approach allocates fewer data points for prior learning early on while reserving more for later refinement, enabling progressively more targeted fine-tuning.

Like Wu et al. (2024), we apply a relaxation of the PAC-Bayes-kl bound in our experiments by optimizing the McAllester bound (McAllester, 1998), while the bound evaluation is conducted using the split kl-based bound. We always utilize a set of factorized Gaussian distributions to represent the priors and posteriors associated with every trainable parameter within the classifiers. These distributions take the form $\pi = \mathcal{N}(w, \sigma I)$, where $w \in \mathbb{R}^d$ is the mean vector and $\sigma$ denotes the scalar variance parameter. Initially, an uninformed prior $\pi_0 = \mathcal{N}(w_0, \sigma_0 I)$ is employed, which does not depend on the training data. Here, the mean parameter is randomly initialized using Kaiming uniform initialization, while the log-variance parameter is initialized to a fixed value of (-4.6). We train all models using the hyperparameters provided in Table 1, specified with the header 'PAC-Bayes bound fitting'.

**Mitigate sample correlation.**  To reduce overfitting caused by high correlation among consecutive samples—where states are highly similar and their associated target values nearly identical—a thinning strategy is employed. This strategy selectively samples fewer data points to weaken temporal dependence. Specifically, in environments such as `Ant` and `Half-Cheetah`, which tend to produce longer episodes, every fifth sample is retained. In contrast, for environments with shorter episodes, such as `Humanoid`, `Hopper`, and `Walker2d`, every third sample is retained. This approach reduces redundancy while ensuring that the dataset still contains sufficient information for training.

**Effect of the local reparameterization trick**  We incorporate the Local Reparameterization Trick (LRT) into our variational Bayesian neural network architecture, where, to our knowledge, it has not been used in previous Bayesian approaches to reinforcement learning. Unlike standard sampling methods that draw one set of weights per layer and propagate forward, LRT enables sampling at the level of individual pre-activations, which significantly reduces the variance of gradient estimates during training. This, in turn, leads to more stable optimization and an improved posterior. To evaluate its impact, we compare the tightness of all our baselines on the Humanoid environment for two versions of our model, with and without LRT. As shown in

Table 1: *Hyperparameters used in our experiments.* This includes both hyperparameters applied to the Actor-Critic setup in agent training and Bayesian model training for PAC-Bayes bound fitting.

| **Agent training:** | |
| --- | --- |
| Evaluation episodes (evaluation mode) | 1 |
| Evaluation frequency (evaluation mode) | 1 |
| Evaluation episodes (training mode) | - |
| Evaluation frequency (training mode) | - |
| Discount factor ($\gamma$) | 0.99 |
| $n$-step returns | 1 step |
| Replay ratio | 1 |
| Number of critic networks | 10 |
| Replay buffer size | 100,000 |
| Maximum timesteps* | 300,000 |
| Number of hidden layers for all networks | 2 |
| Number of hidden units per layer | 256 |
| Nonlinearity | CReLU |
| Mini-batch size | 256 |
| Network regularization method | Layer normalization (LN) (Ball et al., 2023) |
| Actor/critic optimizer | Adam (Kingma & Ba, 2015) |
| Optimizer learning rates ($\eta_\phi, \eta_\theta$) | 3e-4 |
| Polyak averaging parameter ($\tau$) | 5e-3 |
| **PAC-Bayes bound fitting:** | |
| Number of hidden layers | 2 |
| Number of hidden units per layer | 256 |
| Nonlinearity | ReLU |
| Network regularization method | Layer normalization (LN) (Ball et al., 2023) |
| BNN optimizer | Adam (Kingma & Ba, 2015) |
| Optimizer learning rate | 2e-2 |
| Gradient clipping type | max-norm |
| Gradient clipping threshold | 1 |
| Learning rate scheduler | StepLR |
| Learning rate decay factor | 0.5 |
| Scheduler step size (epochs) | 10 |

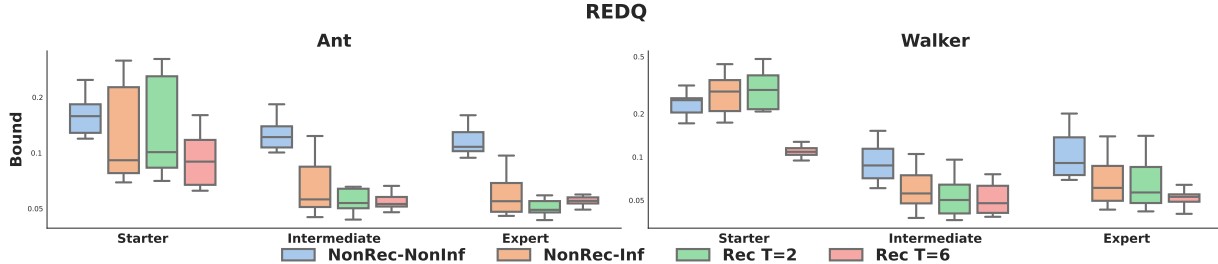

Figure 14: *Bound values.* Normalized bound values for REDQ on two MuJoCo environments and all policy quality levels. Results are aggregated across all seeds and repetitions.

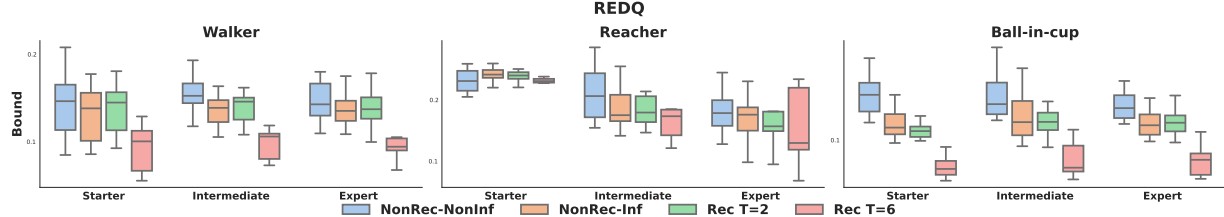

Figure 15: *Bound values.* Normalized bound values for REDQ on three DM Control environments and all policy quality levels. Results are aggregated across all seeds and repetitions.

Figure 13, applying LRT consistently yields tighter bounds. Results are aggregated over five policy instances and five repetitions each, considering only the expert-level policy. In both cases, the results show a clear improvement. These expert-policy improvements suggest gains at other policy levels as well.

### D.5 Computational requirements

Experiments were conducted on a single computer equipped with a GeForce RTX 4090 GPU, an Intel(R) Core(TM) i7-14700K CPU (5.6 GHz), and 96 GB of memory. Training five policy instances for REDQ to convergence in each environment requires approximately 30 minutes per instance, totaling 150 minutes. Collecting validation and test episodes requires around 20 minutes per policy level, totaling 60 minutes. Model training and PAC-Bayesian bound computation across five policy instances, five repetitions, four baselines, and three policies take four minutes per run, totaling approximately 1200 minutes per environment.

## E    Further results

### E.1    Additional REDQ Bounds

We evaluate REDQ on two additional MuJoCo environments (Figure 14), as well as on two further benchmarks-DM Control (Figure 15) and Meta-World (Figure 16)-and observe consistent qualitative results.

### E.2    Correlation plots for PPO and SAC

The corresponding plots to Figure 2 for PPO and SAC are provided in Figure 17 and Figure 18, respectively, which exhibit a similar correlation structure.

### E.3    Validation data size

Figure 19 examines the Pearson correlation between normalized bound scores and normalized test errors across policy levels and varying validation data sizes for the Humanoid environment—our most challenging setting due to its high-dimensional state and action spaces. Each heatmap displays a point estimate of these

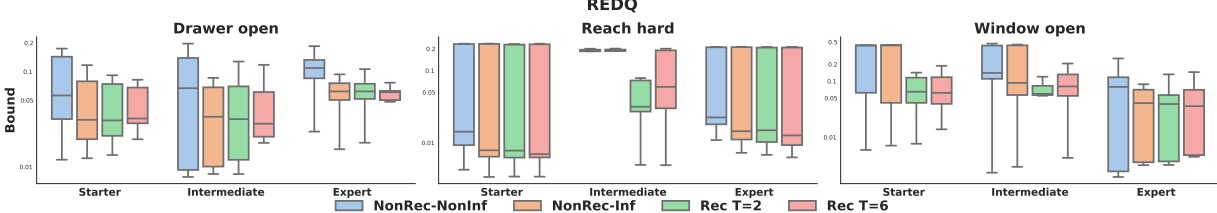

Figure 16: *Bound values.* Normalized bound values for REDQ on three Meta-world environments and all policy quality levels. Results are aggregated across all seeds and repetitions.

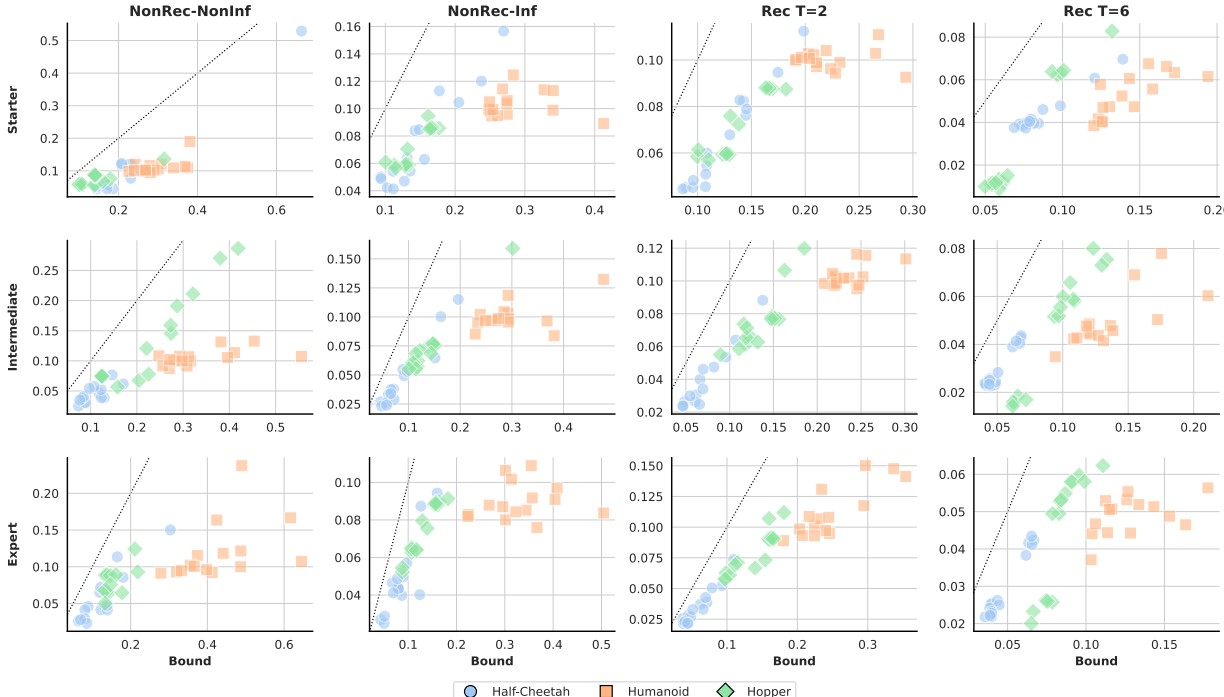

Figure 17: *Correlation between bounds and test errors.* PAC-Bayes bounds (x-axis) are plotted axis against true test errors (y-axis) for PPO across three MuJoCo environments, policy instances, and repetitions to visualize correlation. We observe a high correlation, especially as policies improve and bounds become recursive.

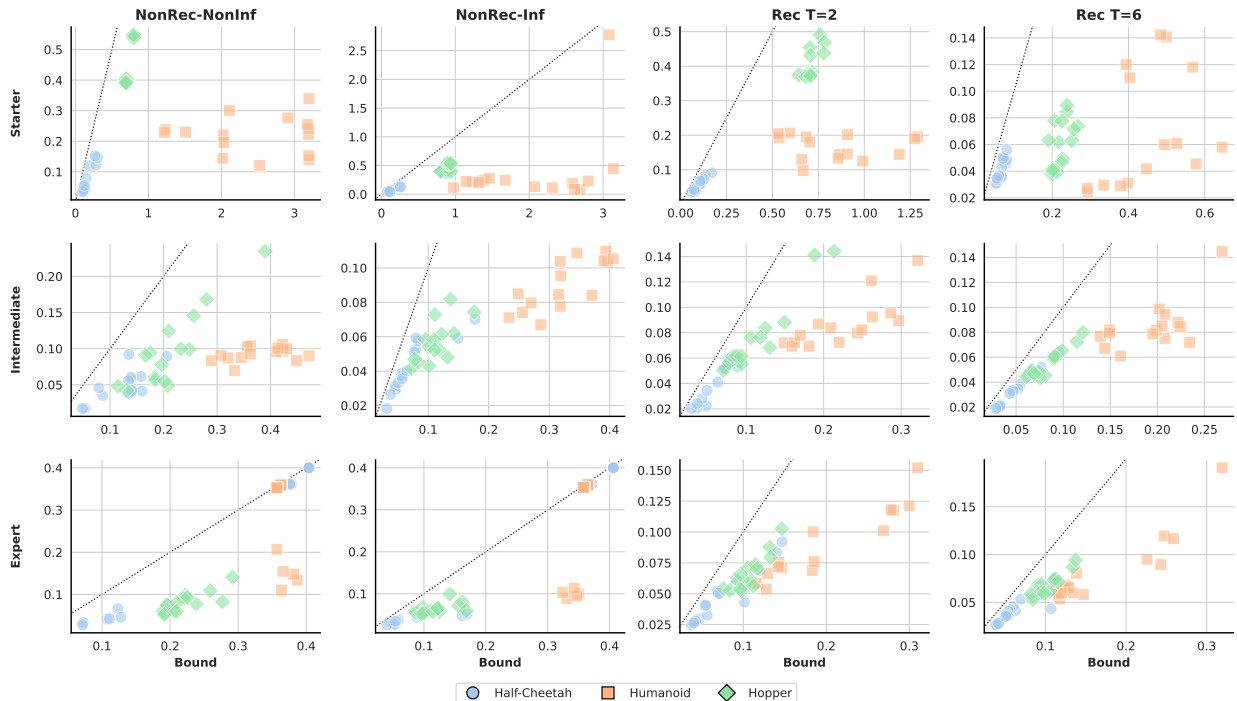

Figure 18: *Correlation between bounds and test errors.* PAC-Bayes bounds (x-axis) are plotted axis against true test errors (y-axis) for SAC across three MuJoCo environments, policy instances, and repetitions to visualize correlation. We observe a high correlation, especially as policies improve and bounds become recursive.

**Pearson Correlation Between Bounds and Test Errors**

Figure 19: *Pearson correlation.* Correlation between bound scores and test error considering five policy instances and five repetitions for each, across various validation sizes and policy levels.

correlations for all baselines across different validation set sizes for a specific policy, aggregated over policy instances and repetitions. Although correlations are generally weaker and more variable under the Starter policy, they tend to improve with stronger policies and larger validation sets for intermediate- and expert-level policies. Recursive bounds, especially those with greater recursion depth, consistently show stronger positive correlations. These findings emphasize that both policy strength and validation data size affect the bound's ability to predict generalization, with recursion providing the most reliable alignment between bounds and test errors.

Table 2: Normalized train error, test error, and bound for REDQ on MuJoCo's walker2d

(a) Starter policy

| Method | Seed 01 | | | Seed 02 | | | Seed 03 | | | Seed 04 | | | Seed 05 | | |
|---|---|---|---|---|---|---|---|---|---|---|---|---|---|---|---|
| | Train | Test | Bound | Train | Test | Bound | Train | Test | Bound | Train | Test | Bound | Train | Test | Bound |
| NonRec-NonInf | 0.150 | 0.142 | 0.292 | 0.103 | 0.103 | 0.173 | 0.182 | 0.186 | 0.252 | 0.158 | 0.161 | 0.254 | 0.097 | 0.097 | 0.221 |
| NonRec-Inf | 0.283 | 0.272 | 0.416 | 0.104 | 0.104 | 0.208 | 0.183 | 0.187 | 0.286 | 0.236 | 0.240 | 0.341 | 0.096 | 0.095 | 0.194 |
| Rec T=2 | 0.360 | 0.347 | 0.475 | 0.105 | 0.105 | 0.210 | 0.189 | 0.193 | 0.292 | 0.269 | 0.273 | 0.369 | 0.158 | 0.156 | 0.245 |
| Rec T=6 | 0.063 | 0.059 | 0.117 | 0.031 | 0.030 | 0.101 | 0.045 | 0.047 | 0.105 | 0.059 | 0.061 | 0.114 | 0.060 | 0.060 | 0.108 |

(b) Intermediate policy

| Method | Seed 01 | | | Seed 02 | | | Seed 03 | | | Seed 04 | | | Seed 05 | | |
|---|---|---|---|---|---|---|---|---|---|---|---|---|---|---|---|
| | Train | Test | Bound | Train | Test | Bound | Train | Test | Bound | Train | Test | Bound | Train | Test | Bound |
| NonRec-NonInf | 0.046 | 0.046 | 0.083 | 0.056 | 0.056 | 0.086 | 0.046 | 0.046 | 0.083 | 0.054 | 0.054 | 0.095 | 0.071 | 0.069 | 0.134 |
| NonRec-Inf | 0.034 | 0.034 | 0.046 | 0.042 | 0.042 | 0.057 | 0.043 | 0.043 | 0.059 | 0.038 | 0.038 | 0.052 | 0.073 | 0.072 | 0.098 |
| Rec T=2 | 0.037 | 0.037 | 0.048 | 0.032 | 0.032 | 0.042 | 0.033 | 0.033 | 0.044 | 0.042 | 0.042 | 0.056 | 0.063 | 0.061 | 0.084 |
| Rec T=6 | 0.031 | 0.031 | 0.040 | 0.031 | 0.031 | 0.042 | 0.040 | 0.041 | 0.057 | 0.039 | 0.039 | 0.052 | 0.054 | 0.053 | 0.072 |

(c) Expert policy

| Method | Seed 01 | | | Seed 02 | | | Seed 03 | | | Seed 04 | | | Seed 05 | | |
|---|---|---|---|---|---|---|---|---|---|---|---|---|---|---|---|
| | Train | Test | Bound | Train | Test | Bound | Train | Test | Bound | Train | Test | Bound | Train | Test | Bound |
| NonRec-NonInf | 0.088 | 0.086 | 0.181 | 0.039 | 0.040 | 0.073 | 0.048 | 0.048 | 0.090 | 0.051 | 0.051 | 0.092 | 0.071 | 0.071 | 0.119 |
| NonRec-Inf | 0.090 | 0.091 | 0.130 | 0.045 | 0.047 | 0.060 | 0.039 | 0.039 | 0.052 | 0.056 | 0.056 | 0.075 | 0.046 | 0.046 | 0.061 |
| Rec T=2 | 0.084 | 0.084 | 0.114 | 0.039 | 0.040 | 0.049 | 0.040 | 0.040 | 0.051 | 0.043 | 0.042 | 0.054 | 0.064 | 0.064 | 0.079 |
| Rec T=6 | 0.076 | 0.072 | 0.106 | 0.034 | 0.035 | 0.043 | 0.040 | 0.040 | 0.050 | 0.041 | 0.041 | 0.053 | 0.043 | 0.043 | 0.053 |

Table 3: Normalized train error, test error, and bound for REDQ on MuJoCo's ant

(a) Starter policy

| Method | Seed 01 | | | Seed 02 | | | Seed 03 | | | Seed 04 | | | Seed 05 | | |
|---|---|---|---|---|---|---|---|---|---|---|---|---|---|---|---|
| | Train | Test | Bound | Train | Test | Bound | Train | Test | Bound | Train | Test | Bound | Train | Test | Bound |
| NonRec-NonInf | 0.074 | 0.070 | 0.143 | 0.061 | 0.067 | 0.147 | 0.112 | 0.085 | 0.177 | 0.070 | 0.068 | 0.124 | 0.195 | 0.156 | 0.281 |
| NonRec-Inf | 0.051 | 0.043 | 0.080 | 0.052 | 0.057 | 0.073 | 0.137 | 0.106 | 0.216 | 0.066 | 0.064 | 0.094 | 0.268 | 0.217 | 0.309 |
| Rec T=2 | 0.060 | 0.053 | 0.085 | 0.056 | 0.060 | 0.079 | 0.198 | 0.160 | 0.256 | 0.072 | 0.071 | 0.101 | 0.283 | 0.230 | 0.320 |
| Rec T=6 | 0.049 | 0.040 | 0.068 | 0.048 | 0.055 | 0.065 | 0.078 | 0.061 | 0.115 | 0.063 | 0.061 | 0.088 | 0.103 | 0.086 | 0.139 |

(b) Intermediate policy

| Method | Seed 01 | | | Seed 02 | | | Seed 03 | | | Seed 04 | | | Seed 05 | | |
|---|---|---|---|---|---|---|---|---|---|---|---|---|---|---|---|
| | Train | Test | Bound | Train | Test | Bound | Train | Test | Bound | Train | Test | Bound | Train | Test | Bound |
| NonRec-NonInf | 0.057 | 0.055 | 0.129 | 0.070 | 0.089 | 0.149 | 0.046 | 0.056 | 0.116 | 0.046 | 0.049 | 0.113 | 0.055 | 0.065 | 0.123 |
| NonRec-Inf | 0.053 | 0.051 | 0.066 | 0.069 | 0.086 | 0.107 | 0.051 | 0.059 | 0.069 | 0.036 | 0.042 | 0.052 | 0.038 | 0.047 | 0.054 |
| Rec T=2 | 0.044 | 0.042 | 0.053 | 0.065 | 0.081 | 0.098 | 0.043 | 0.050 | 0.056 | 0.038 | 0.043 | 0.055 | 0.034 | 0.044 | 0.047 |
| Rec T=6 | 0.039 | 0.038 | 0.052 | 0.046 | 0.049 | 0.064 | 0.039 | 0.041 | 0.052 | 0.035 | 0.040 | 0.051 | 0.041 | 0.043 | 0.054 |

(c) Expert policy

| Method | Seed 01 | | | Seed 02 | | | Seed 03 | | | Seed 04 | | | Seed 05 | | |
|---|---|---|---|---|---|---|---|---|---|---|---|---|---|---|---|
| | Train | Test | Bound | Train | Test | Bound | Train | Test | Bound | Train | Test | Bound | Train | Test | Bound |
| NonRec-NonInf | 0.039 | 0.040 | 0.101 | 0.053 | 0.057 | 0.117 | 0.042 | 0.050 | 0.106 | 0.058 | 0.056 | 0.135 | 0.052 | 0.052 | 0.116 |
| NonRec-Inf | 0.041 | 0.043 | 0.057 | 0.046 | 0.050 | 0.059 | 0.039 | 0.047 | 0.049 | 0.053 | 0.050 | 0.081 | 0.037 | 0.036 | 0.052 |
| Rec T=2 | 0.035 | 0.037 | 0.050 | 0.041 | 0.045 | 0.049 | 0.039 | 0.047 | 0.049 | 0.069 | 0.068 | 0.098 | 0.038 | 0.037 | 0.049 |
| Rec T=6 | 0.040 | 0.040 | 0.053 | 0.043 | 0.044 | 0.057 | 0.044 | 0.043 | 0.057 | 0.040 | 0.038 | 0.059 | 0.040 | 0.040 | 0.052 |

## E.4 Raw numbers

We include the raw individual values averaged over five repetitions in all scenarios.

Table 4: Normalized train error, test error, and bound for REDQ on MuJoCo's cheetah

(a) Starter policy

| | Seed 01 | | | Seed 02 | | | Seed 03 | | | Seed 04 | | | Seed 05 | | |
|---|---|---|---|---|---|---|---|---|---|---|---|---|---|---|---|
| Method | Train | Test | Bound | Train | Test | Bound | Train | Test | Bound | Train | Test | Bound | Train | Test | Bound |
| NonRec-NonInf | 0.071 | 0.072 | 0.176 | 0.058 | 0.058 | 0.135 | 0.062 | 0.062 | 0.128 | 0.066 | 0.065 | 0.153 | 0.060 | 0.061 | 0.148 |
| NonRec-Inf | 0.043 | 0.043 | 0.066 | 0.039 | 0.039 | 0.063 | 0.037 | 0.037 | 0.059 | 0.047 | 0.047 | 0.070 | 0.042 | 0.042 | 0.066 |
| Rec T=2 | 0.042 | 0.042 | 0.066 | 0.044 | 0.044 | 0.066 | 0.037 | 0.037 | 0.057 | 0.051 | 0.050 | 0.076 | 0.039 | 0.039 | 0.060 |
| Rec T=6 | 0.056 | 0.057 | 0.081 | 0.037 | 0.038 | 0.059 | 0.032 | 0.032 | 0.053 | 0.050 | 0.049 | 0.074 | 0.041 | 0.041 | 0.062 |

(b) Intermediate policy

| | Seed 01 | | | Seed 02 | | | Seed 03 | | | Seed 04 | | | Seed 05 | | |
|---|---|---|---|---|---|---|---|---|---|---|---|---|---|---|---|
| Method | Train | Test | Bound | Train | Test | Bound | Train | Test | Bound | Train | Test | Bound | Train | Test | Bound |
| NonRec-NonInf | 0.066 | 0.066 | 0.149 | 0.068 | 0.067 | 0.152 | 0.033 | 0.033 | 0.093 | 0.044 | 0.044 | 0.117 | 0.062 | 0.062 | 0.148 |
| NonRec-Inf | 0.042 | 0.043 | 0.060 | 0.045 | 0.045 | 0.063 | 0.030 | 0.030 | 0.044 | 0.040 | 0.040 | 0.058 | 0.043 | 0.043 | 0.063 |
| Rec T=2 | 0.047 | 0.047 | 0.064 | 0.041 | 0.041 | 0.057 | 0.029 | 0.029 | 0.042 | 0.039 | 0.039 | 0.057 | 0.046 | 0.046 | 0.064 |
| Rec T=6 | 0.046 | 0.046 | 0.064 | 0.046 | 0.046 | 0.064 | 0.032 | 0.032 | 0.046 | 0.039 | 0.039 | 0.054 | 0.046 | 0.046 | 0.062 |

(c) Expert policy

| | Seed 01 | | | Seed 02 | | | Seed 03 | | | Seed 04 | | | Seed 05 | | |
|---|---|---|---|---|---|---|---|---|---|---|---|---|---|---|---|
| Method | Train | Test | Bound | Train | Test | Bound | Train | Test | Bound | Train | Test | Bound | Train | Test | Bound |
| NonRec-NonInf | 0.045 | 0.045 | 0.107 | 0.050 | 0.052 | 0.124 | 0.029 | 0.029 | 0.070 | 0.063 | 0.058 | 0.136 | 0.061 | 0.065 | 0.140 |
| NonRec-Inf | 0.033 | 0.033 | 0.046 | 0.043 | 0.046 | 0.063 | 0.018 | 0.018 | 0.027 | 0.041 | 0.036 | 0.050 | 0.045 | 0.049 | 0.061 |
| Rec T=2 | 0.040 | 0.040 | 0.055 | 0.043 | 0.046 | 0.060 | 0.018 | 0.018 | 0.026 | 0.044 | 0.038 | 0.051 | 0.041 | 0.045 | 0.054 |
| Rec T=6 | 0.037 | 0.037 | 0.048 | 0.045 | 0.046 | 0.059 | 0.021 | 0.021 | 0.030 | 0.042 | 0.042 | 0.055 | 0.046 | 0.047 | 0.059 |

Table 5: Normalized train error, test error, and bound for REDQ on MuJoCo's humanoid

(a) Starter policy

| | Seed 01 | | | Seed 02 | | | Seed 03 | | | Seed 04 | | | Seed 05 | | |
|---|---|---|---|---|---|---|---|---|---|---|---|---|---|---|---|
| Method | Train | Test | Bound | Train | Test | Bound | Train | Test | Bound | Train | Test | Bound | Train | Test | Bound |
| NonRec-NonInf | 0.159 | 0.157 | 0.554 | 0.125 | 0.129 | 0.291 | 0.052 | 0.053 | 0.954 | 0.171 | 0.169 | 0.405 | 0.176 | 0.171 | 0.502 |
| NonRec-Inf | 0.143 | 0.142 | 0.543 | 0.108 | 0.108 | 0.279 | 0.044 | 0.043 | 0.530 | 0.160 | 0.161 | 0.393 | 0.167 | 0.161 | 0.382 |
| Rec T=2 | 0.147 | 0.144 | 0.365 | 0.125 | 0.126 | 0.292 | 0.051 | 0.053 | 0.428 | 0.166 | 0.166 | 0.332 | 0.172 | 0.170 | 0.350 |
| Rec T=6 | 0.115 | 0.112 | 0.289 | 0.092 | 0.090 | 0.216 | 0.018 | 0.019 | 0.191 | 0.117 | 0.119 | 0.255 | 0.142 | 0.139 | 0.306 |

(b) Intermediate policy

| | Seed 01 | | | Seed 02 | | | Seed 03 | | | Seed 04 | | | Seed 05 | | |
|---|---|---|---|---|---|---|---|---|---|---|---|---|---|---|---|
| Method | Train | Test | Bound | Train | Test | Bound | Train | Test | Bound | Train | Test | Bound | Train | Test | Bound |
| NonRec-NonInf | 0.151 | 0.157 | 0.418 | 0.103 | 0.105 | 0.241 | 0.066 | 0.066 | 0.130 | 0.110 | 0.112 | 0.322 | 0.116 | 0.122 | 0.343 |
| NonRec-Inf | 0.133 | 0.137 | 0.366 | 0.101 | 0.104 | 0.201 | 0.064 | 0.065 | 0.131 | 0.110 | 0.111 | 0.260 | 0.106 | 0.110 | 0.310 |
| Rec T=2 | 0.150 | 0.153 | 0.276 | 0.107 | 0.110 | 0.171 | 0.065 | 0.066 | 0.113 | 0.115 | 0.117 | 0.224 | 0.117 | 0.119 | 0.226 |
| Rec T=6 | 0.117 | 0.120 | 0.227 | 0.080 | 0.084 | 0.147 | 0.046 | 0.049 | 0.092 | 0.094 | 0.096 | 0.172 | 0.082 | 0.086 | 0.161 |

(c) Expert policy

| | Seed 01 | | | Seed 02 | | | Seed 03 | | | Seed 04 | | | Seed 05 | | |
|---|---|---|---|---|---|---|---|---|---|---|---|---|---|---|---|
| Method | Train | Test | Bound | Train | Test | Bound | Train | Test | Bound | Train | Test | Bound | Train | Test | Bound |
| NonRec-NonInf | 0.095 | 0.098 | 0.268 | 0.073 | 0.071 | 0.177 | 0.081 | 0.088 | 0.244 | 0.082 | 0.083 | 0.261 | 0.106 | 0.102 | 0.284 |
| NonRec-Inf | 0.104 | 0.107 | 0.229 | 0.067 | 0.066 | 0.153 | 0.062 | 0.066 | 0.188 | 0.085 | 0.086 | 0.217 | 0.101 | 0.098 | 0.241 |
| Rec T=2 | 0.109 | 0.114 | 0.193 | 0.072 | 0.071 | 0.121 | 0.074 | 0.077 | 0.153 | 0.115 | 0.115 | 0.223 | 0.135 | 0.128 | 0.238 |
| Rec T=6 | 0.102 | 0.107 | 0.176 | 0.067 | 0.067 | 0.110 | 0.076 | 0.078 | 0.147 | 0.091 | 0.092 | 0.175 | 0.101 | 0.099 | 0.190 |

Table 6: Normalized train error, test error, and bound for REDQ on MuJoCo's hopper

(a) Starter policy

| | SEED 01 | | | SEED 02 | | | SEED 03 | | | SEED 04 | | | SEED 05 | | |
| METHOD | Train | Test | Bound | Train | Test | Bound | Train | Test | Bound | Train | Test | Bound | Train | Test | Bound |
| --- | --- | --- | --- | --- | --- | --- | --- | --- | --- | --- | --- | --- | --- | --- | --- |
| NonRec-NonInf | 0.567 | 0.569 | 0.759 | 0.341 | 0.320 | 0.868 | 0.007 | 0.007 | 0.269 | 0.201 | 0.187 | 1.011 | 0.122 | 0.122 | 0.218 |
| NonRec-Inf | 0.575 | 0.577 | 0.854 | 0.347 | 0.326 | 1.019 | 0.009 | 0.009 | 0.283 | 0.200 | 0.186 | 1.324 | 0.189 | 0.189 | 0.314 |
| Rec T=2 | 0.580 | 0.583 | 0.856 | 0.346 | 0.326 | 1.014 | 0.009 | 0.009 | 0.282 | 0.200 | 0.187 | 1.260 | 0.261 | 0.261 | 0.389 |
| Rec T=6 | 0.156 | 0.156 | 0.321 | 0.243 | 0.228 | 0.803 | 0.004 | 0.004 | 0.399 | 0.166 | 0.155 | 0.859 | 0.058 | 0.058 | 0.123 |

(b) Intermediate policy

| | SEED 01 | | | SEED 02 | | | SEED 03 | | | SEED 04 | | | SEED 05 | | |
| METHOD | Train | Test | Bound | Train | Test | Bound | Train | Test | Bound | Train | Test | Bound | Train | Test | Bound |
| --- | --- | --- | --- | --- | --- | --- | --- | --- | --- | --- | --- | --- | --- | --- | --- |
| NonRec-NonInf | 0.079 | 0.081 | 0.181 | 0.075 | 0.076 | 0.165 | 0.073 | 0.074 | 0.197 | 0.096 | 0.095 | 0.233 | 0.060 | 0.060 | 0.155 |
| NonRec-Inf | 0.069 | 0.070 | 0.109 | 0.059 | 0.058 | 0.087 | 0.070 | 0.070 | 0.117 | 0.115 | 0.116 | 0.260 | 0.062 | 0.062 | 0.096 |
| Rec T=2 | 0.073 | 0.074 | 0.111 | 0.062 | 0.062 | 0.093 | 0.069 | 0.069 | 0.110 | 0.234 | 0.233 | 0.348 | 0.061 | 0.061 | 0.094 |
| Rec T=6 | 0.064 | 0.063 | 0.094 | 0.054 | 0.053 | 0.081 | 0.061 | 0.061 | 0.096 | 0.071 | 0.071 | 0.115 | 0.059 | 0.059 | 0.092 |

(c) Expert policy

| | SEED 01 | | | SEED 02 | | | SEED 03 | | | SEED 04 | | | SEED 05 | | |
| METHOD | Train | Test | Bound | Train | Test | Bound | Train | Test | Bound | Train | Test | Bound | Train | Test | Bound |
| --- | --- | --- | --- | --- | --- | --- | --- | --- | --- | --- | --- | --- | --- | --- | --- |
| NonRec-NonInf | 0.072 | 0.071 | 0.184 | 0.121 | 0.121 | 0.241 | 0.053 | 0.053 | 0.100 | 0.095 | 0.088 | 0.207 | 0.094 | 0.092 | 0.210 |
| NonRec-Inf | 0.058 | 0.057 | 0.089 | 0.060 | 0.061 | 0.091 | 0.046 | 0.046 | 0.068 | 0.106 | 0.098 | 0.244 | 0.060 | 0.059 | 0.095 |
| Rec T=2 | 0.058 | 0.058 | 0.090 | 0.062 | 0.062 | 0.090 | 0.044 | 0.044 | 0.060 | 0.196 | 0.181 | 0.305 | 0.066 | 0.065 | 0.096 |
| Rec T=6 | 0.056 | 0.056 | 0.086 | 0.061 | 0.061 | 0.086 | 0.040 | 0.041 | 0.054 | 0.058 | 0.058 | 0.095 | 0.057 | 0.056 | 0.084 |

Table 7: Normalized train error, test error, and bound for SAC on MuJoCo's cheetah

(a) Starter policy

| | SEED 01 | | | SEED 02 | | | SEED 03 | | |
| METHOD | Train | Test | Bound | Train | Test | Bound | Train | Test | Bound |
| --- | --- | --- | --- | --- | --- | --- | --- | --- | --- |
| NonRec-NonInf | 0.066 | 0.066 | 0.137 | 0.062 | 0.062 | 0.148 | 0.086 | 0.086 | 0.183 |
| NonRec-Inf | 0.077 | 0.077 | 0.140 | 0.050 | 0.050 | 0.096 | 0.069 | 0.069 | 0.139 |
| Rec T=2 | 0.059 | 0.059 | 0.107 | 0.063 | 0.063 | 0.112 | 0.051 | 0.051 | 0.088 |
| Rec T=6 | 0.036 | 0.036 | 0.063 | 0.048 | 0.049 | 0.074 | 0.035 | 0.035 | 0.061 |

(b) Intermediate policy

| | SEED 01 | | | SEED 02 | | | SEED 03 | | |
| METHOD | Train | Test | Bound | Train | Test | Bound | Train | Test | Bound |
| --- | --- | --- | --- | --- | --- | --- | --- | --- | --- |
| NonRec-NonInf | 0.042 | 0.042 | 0.080 | 0.064 | 0.065 | 0.155 | 0.038 | 0.040 | 0.143 |
| NonRec-Inf | 0.029 | 0.029 | 0.044 | 0.057 | 0.058 | 0.113 | 0.038 | 0.041 | 0.062 |
| Rec T=2 | 0.023 | 0.023 | 0.039 | 0.055 | 0.056 | 0.079 | 0.046 | 0.048 | 0.068 |
| Rec T=6 | 0.020 | 0.020 | 0.031 | 0.051 | 0.051 | 0.073 | 0.031 | 0.034 | 0.049 |

(c) Expert policy

| | SEED 01 | | | SEED 02 | | | SEED 03 | | |
| METHOD | Train | Test | Bound | Train | Test | Bound | Train | Test | Bound |
| --- | --- | --- | --- | --- | --- | --- | --- | --- | --- |
| NonRec-NonInf | 0.042 | 0.042 | 0.097 | 0.401 | 0.400 | 0.404 | 0.305 | 0.298 | 0.327 |
| NonRec-Inf | 0.034 | 0.034 | 0.059 | 0.401 | 0.400 | 0.406 | 0.109 | 0.108 | 0.169 |
| Rec T=2 | 0.028 | 0.028 | 0.045 | 0.066 | 0.066 | 0.113 | 0.045 | 0.051 | 0.071 |
| Rec T=6 | 0.030 | 0.030 | 0.045 | 0.049 | 0.049 | 0.079 | 0.037 | 0.037 | 0.053 |

Table 8: Normalized train error, test error, and bound for SAC on MuJoCo's humanoid

(a) Starter policy

| | Seed 01 | | | Seed 02 | | | Seed 03 | | |
|---|---|---|---|---|---|---|---|---|---|
| Method | Train | Test | Bound | Train | Test | Bound | Train | Test | Bound |
| NonRec-NonInf | 0.248 | 0.219 | 3.198 | 0.240 | 0.243 | 1.620 | 0.194 | 0.198 | 2.533 |
| NonRec-Inf | 0.167 | 0.145 | 2.051 | 0.741 | 0.740 | 1.648 | 0.219 | 0.233 | 2.400 |
| Rec T=2 | 0.150 | 0.151 | 1.049 | 0.202 | 0.196 | 0.610 | 0.144 | 0.151 | 0.873 |
| Rec T=6 | 0.066 | 0.053 | 0.538 | 0.133 | 0.126 | 0.470 | 0.028 | 0.028 | 0.340 |

(b) Intermediate policy

| | Seed 01 | | | Seed 02 | | | Seed 03 | | |
|---|---|---|---|---|---|---|---|---|---|
| Method | Train | Test | Bound | Train | Test | Bound | Train | Test | Bound |
| NonRec-NonInf | 0.087 | 0.089 | 0.332 | 0.095 | 0.094 | 0.435 | 0.093 | 0.094 | 0.372 |
| NonRec-Inf | 0.077 | 0.080 | 0.278 | 0.101 | 0.101 | 0.391 | 0.088 | 0.089 | 0.304 |
| Rec T=2 | 0.099 | 0.100 | 0.241 | 0.089 | 0.088 | 0.268 | 0.071 | 0.072 | 0.164 |
| Rec T=6 | 0.083 | 0.084 | 0.203 | 0.090 | 0.090 | 0.207 | 0.079 | 0.080 | 0.171 |

(c) Expert policy

| | Seed 01 | | | Seed 02 | | | Seed 03 | | |
|---|---|---|---|---|---|---|---|---|---|
| Method | Train | Test | Bound | Train | Test | Bound | Train | Test | Bound |
| NonRec-NonInf | 0.360 | 0.360 | 0.363 | 0.152 | 0.151 | 0.371 | 0.353 | 0.353 | 0.357 |
| NonRec-Inf | 0.360 | 0.360 | 0.365 | 0.103 | 0.100 | 0.339 | 0.353 | 0.353 | 0.358 |
| Rec T=2 | 0.061 | 0.061 | 0.127 | 0.127 | 0.122 | 0.288 | 0.078 | 0.078 | 0.168 |
| Rec T=6 | 0.062 | 0.062 | 0.127 | 0.126 | 0.122 | 0.259 | 0.065 | 0.065 | 0.128 |

Table 9: Normalized train error, test error, and bound for SAC on MuJoCo's hopper

(a) Starter policy

| Method | Seed 01 | | | Seed 02 | | | Seed 03 | | |
|---|---|---|---|---|---|---|---|---|---|
| | Train | Test | Bound | Train | Test | Bound | Train | Test | Bound |
| NonRec-NonInf | 0.399 | 0.402 | 0.700 | 0.392 | 0.391 | 0.692 | 0.543 | 0.542 | 0.799 |
| NonRec-Inf | 0.404 | 0.407 | 0.832 | 0.398 | 0.398 | 0.880 | 0.540 | 0.540 | 0.915 |
| Rec T=2 | 0.379 | 0.380 | 0.700 | 0.374 | 0.373 | 0.687 | 0.456 | 0.457 | 0.747 |
| Rec T=6 | 0.064 | 0.064 | 0.244 | 0.041 | 0.041 | 0.208 | 0.078 | 0.079 | 0.220 |

(b) Intermediate policy

| Method | Seed 01 | | | Seed 02 | | | Seed 03 | | |
|---|---|---|---|---|---|---|---|---|---|
| | Train | Test | Bound | Train | Test | Bound | Train | Test | Bound |
| NonRec-NonInf | 0.100 | 0.100 | 0.243 | 0.080 | 0.080 | 0.159 | 0.109 | 0.109 | 0.234 |
| NonRec-Inf | 0.065 | 0.066 | 0.142 | 0.055 | 0.054 | 0.100 | 0.052 | 0.051 | 0.097 |
| Rec T=2 | 0.072 | 0.072 | 0.117 | 0.076 | 0.077 | 0.105 | 0.080 | 0.080 | 0.121 |
| Rec T=6 | 0.062 | 0.063 | 0.097 | 0.048 | 0.047 | 0.065 | 0.053 | 0.053 | 0.084 |

(c) Expert policy

| Method | Seed 01 | | | Seed 02 | | | Seed 03 | | |
|---|---|---|---|---|---|---|---|---|---|
| | Train | Test | Bound | Train | Test | Bound | Train | Test | Bound |
| NonRec-NonInf | 0.072 | 0.074 | 0.208 | 0.099 | 0.090 | 0.225 | 0.081 | 0.078 | 0.232 |
| NonRec-Inf | 0.067 | 0.068 | 0.104 | 0.060 | 0.057 | 0.115 | 0.068 | 0.066 | 0.138 |
| Rec T=2 | 0.072 | 0.073 | 0.110 | 0.057 | 0.055 | 0.100 | 0.073 | 0.072 | 0.113 |
| Rec T=6 | 0.061 | 0.063 | 0.088 | 0.063 | 0.062 | 0.099 | 0.080 | 0.080 | 0.122 |

Table 10: Normalized train error, test error, and bound for PPO on MuJoCo's Cheetah

(a) Starter policy

| Method | Seed 01 | | | Seed 02 | | | Seed 03 | | |
|---|---|---|---|---|---|---|---|---|---|
| | Train | Test | Bound | Train | Test | Bound | Train | Test | Bound |
| NonRec-NonInf | 0.166 | 0.166 | 0.277 | 0.074 | 0.074 | 0.185 | 0.087 | 0.086 | 0.198 |
| NonRec-Inf | 0.057 | 0.057 | 0.125 | 0.090 | 0.090 | 0.160 | 0.078 | 0.078 | 0.163 |
| Rec T=2 | 0.063 | 0.063 | 0.119 | 0.055 | 0.054 | 0.111 | 0.080 | 0.080 | 0.146 |
| Rec T=6 | 0.047 | 0.047 | 0.094 | 0.040 | 0.040 | 0.076 | 0.045 | 0.045 | 0.088 |

(b) Intermediate policy

| Method | Seed 01 | | | Seed 02 | | | Seed 03 | | |
|---|---|---|---|---|---|---|---|---|---|
| | Train | Test | Bound | Train | Test | Bound | Train | Test | Bound |
| NonRec-NonInf | 0.054 | 0.053 | 0.137 | 0.048 | 0.048 | 0.103 | 0.040 | 0.040 | 0.086 |
| NonRec-Inf | 0.057 | 0.056 | 0.096 | 0.030 | 0.030 | 0.060 | 0.056 | 0.056 | 0.110 |
| Rec T=2 | 0.064 | 0.063 | 0.108 | 0.034 | 0.034 | 0.064 | 0.030 | 0.030 | 0.059 |
| Rec T=6 | 0.042 | 0.041 | 0.066 | 0.023 | 0.023 | 0.043 | 0.025 | 0.025 | 0.046 |

(c) Expert policy

| Method | Seed 01 | | | Seed 02 | | | Seed 03 | | |
|---|---|---|---|---|---|---|---|---|---|
| | Train | Test | Bound | Train | Test | Bound | Train | Test | Bound |
| NonRec-NonInf | 0.065 | 0.065 | 0.167 | 0.063 | 0.065 | 0.112 | 0.041 | 0.041 | 0.099 |
| NonRec-Inf | 0.052 | 0.052 | 0.106 | 0.042 | 0.045 | 0.075 | 0.046 | 0.046 | 0.075 |
| Rec T=2 | 0.049 | 0.049 | 0.081 | 0.026 | 0.029 | 0.050 | 0.028 | 0.028 | 0.049 |
| Rec T=6 | 0.041 | 0.041 | 0.065 | 0.023 | 0.025 | 0.041 | 0.022 | 0.022 | 0.039 |

Table 11: Normalized train error, test error, and bound for PPO on MuJoCo's Humanoid

(a) Starter policy

| METHOD | Seed 01 | | | Seed 02 | | | Seed 03 | | |
|---|---|---|---|---|---|---|---|---|---|
| | Train | Test | Bound | Train | Test | Bound | Train | Test | Bound |
| NonRec-NonInf | 0.101 | 0.108 | 0.262 | 0.112 | 0.108 | 0.324 | 0.121 | 0.121 | 0.299 |
| NonRec-Inf | 0.099 | 0.104 | 0.266 | 0.105 | 0.102 | 0.336 | 0.104 | 0.105 | 0.259 |
| Rec T=2 | 0.097 | 0.103 | 0.212 | 0.099 | 0.097 | 0.249 | 0.101 | 0.100 | 0.208 |
| Rec T=6 | 0.053 | 0.058 | 0.147 | 0.055 | 0.050 | 0.146 | 0.052 | 0.050 | 0.137 |

(b) Intermediate policy

| METHOD | Seed 01 | | | Seed 02 | | | Seed 03 | | |
|---|---|---|---|---|---|---|---|---|---|
| | Train | Test | Bound | Train | Test | Bound | Train | Test | Bound |
| NonRec-NonInf | 0.113 | 0.112 | 0.427 | 0.101 | 0.102 | 0.293 | 0.106 | 0.103 | 0.291 |
| NonRec-Inf | 0.104 | 0.103 | 0.358 | 0.095 | 0.097 | 0.256 | 0.105 | 0.101 | 0.274 |
| Rec T=2 | 0.106 | 0.106 | 0.247 | 0.098 | 0.100 | 0.222 | 0.108 | 0.103 | 0.239 |
| Rec T=6 | 0.045 | 0.045 | 0.122 | 0.042 | 0.043 | 0.115 | 0.065 | 0.061 | 0.170 |

(c) Expert policy

| METHOD | Seed 01 | | | Seed 02 | | | Seed 03 | | |
|---|---|---|---|---|---|---|---|---|---|
| | Train | Test | Bound | Train | Test | Bound | Train | Test | Bound |
| NonRec-NonInf | 0.121 | 0.123 | 0.535 | 0.115 | 0.113 | 0.377 | 0.123 | 0.124 | 0.373 |
| NonRec-Inf | 0.086 | 0.084 | 0.385 | 0.100 | 0.100 | 0.346 | 0.085 | 0.085 | 0.267 |
| Rec T=2 | 0.101 | 0.102 | 0.237 | 0.134 | 0.131 | 0.297 | 0.101 | 0.103 | 0.213 |
| Rec T=6 | 0.050 | 0.049 | 0.153 | 0.051 | 0.053 | 0.123 | 0.045 | 0.045 | 0.108 |

Table 12: Normalized train error, test error, and bound for PPO on MuJoCo's Hopper

(a) Starter policy

| | Seed 01 | | | Seed 02 | | | Seed 03 | | |
|---|---|---|---|---|---|---|---|---|---|
| Method | Train | Test | Bound | Train | Test | Bound | Train | Test | Bound |
| NonRec-NonInf | 0.058 | 0.058 | 0.103 | 0.076 | 0.079 | 0.187 | 0.086 | 0.086 | 0.141 |
| NonRec-Inf | 0.059 | 0.059 | 0.130 | 0.071 | 0.068 | 0.124 | 0.086 | 0.086 | 0.168 |
| Rec T=2 | 0.059 | 0.060 | 0.127 | 0.066 | 0.065 | 0.116 | 0.088 | 0.088 | 0.170 |
| Rec T=6 | 0.011 | 0.011 | 0.057 | 0.062 | 0.067 | 0.104 | 0.013 | 0.012 | 0.057 |

(b) Intermediate policy

| | Seed 01 | | | Seed 02 | | | Seed 03 | | |
|---|---|---|---|---|---|---|---|---|---|
| Method | Train | Test | Bound | Train | Test | Bound | Train | Test | Bound |
| NonRec-NonInf | 0.074 | 0.075 | 0.124 | 0.134 | 0.134 | 0.252 | 0.189 | 0.184 | 0.301 |
| NonRec-Inf | 0.076 | 0.076 | 0.146 | 0.067 | 0.064 | 0.121 | 0.078 | 0.080 | 0.148 |
| Rec T=2 | 0.077 | 0.077 | 0.151 | 0.076 | 0.074 | 0.131 | 0.075 | 0.074 | 0.125 |
| Rec T=6 | 0.016 | 0.016 | 0.065 | 0.061 | 0.059 | 0.108 | 0.066 | 0.067 | 0.112 |

(c) Expert policy

| | Seed 01 | | | Seed 02 | | | Seed 03 | | |
|---|---|---|---|---|---|---|---|---|---|
| Method | Train | Test | Bound | Train | Test | Bound | Train | Test | Bound |
| NonRec-NonInf | 0.088 | 0.088 | 0.137 | 0.071 | 0.075 | 0.144 | 0.082 | 0.082 | 0.178 |
| NonRec-Inf | 0.089 | 0.089 | 0.158 | 0.073 | 0.075 | 0.135 | 0.059 | 0.060 | 0.101 |
| Rec T=2 | 0.091 | 0.091 | 0.163 | 0.082 | 0.086 | 0.150 | 0.062 | 0.062 | 0.102 |
| Rec T=6 | 0.024 | 0.024 | 0.072 | 0.056 | 0.059 | 0.095 | 0.052 | 0.053 | 0.086 |

Table 13: Normalized train error, test error, and bound for REDQ on Meta-world's Drawer open

(a) Starter policy

| | Seed 01 | | | Seed 02 | | | Seed 03 | | | Seed 04 | | | Seed 05 | | |
|---|---|---|---|---|---|---|---|---|---|---|---|---|---|---|---|
| Method | Train | Test | Bound | Train | Test | Bound | Train | Test | Bound | Train | Test | Bound | Train | Test | Bound |
| NonRec-NonInf | 0.023 | 0.023 | 0.040 | 0.070 | 0.070 | 0.160 | 0.062 | 0.062 | 0.138 | 0.008 | 0.008 | 0.014 | 0.025 | 0.025 | 0.052 |
| NonRec-Inf | 0.013 | 0.013 | 0.020 | 0.069 | 0.069 | 0.088 | 0.067 | 0.068 | 0.086 | 0.007 | 0.007 | 0.013 | 0.021 | 0.021 | 0.031 |
| Rec T=2 | 0.014 | 0.014 | 0.022 | 0.063 | 0.063 | 0.082 | 0.055 | 0.055 | 0.071 | 0.009 | 0.009 | 0.015 | 0.023 | 0.023 | 0.033 |
| Rec T=6 | 0.019 | 0.019 | 0.030 | 0.060 | 0.060 | 0.075 | 0.052 | 0.052 | 0.066 | 0.021 | 0.021 | 0.029 | 0.019 | 0.019 | 0.028 |

(b) Intermediate policy

| | Seed 01 | | | Seed 02 | | | Seed 03 | | | Seed 04 | | | Seed 05 | | |
|---|---|---|---|---|---|---|---|---|---|---|---|---|---|---|---|
| Method | Train | Test | Bound | Train | Test | Bound | Train | Test | Bound | Train | Test | Bound | Train | Test | Bound |
| NonRec-NonInf | 0.068 | 0.067 | 0.128 | 0.157 | 0.158 | 0.243 | 0.005 | 0.005 | 0.010 | 0.005 | 0.005 | 0.009 | 0.050 | 0.050 | 0.089 |
| NonRec-Inf | 0.053 | 0.052 | 0.066 | 0.072 | 0.072 | 0.185 | 0.006 | 0.006 | 0.010 | 0.006 | 0.006 | 0.010 | 0.029 | 0.029 | 0.040 |
| Rec T=2 | 0.053 | 0.052 | 0.065 | 0.061 | 0.061 | 0.085 | 0.007 | 0.007 | 0.012 | 0.007 | 0.007 | 0.011 | 0.023 | 0.023 | 0.032 |
| Rec T=6 | 0.049 | 0.048 | 0.061 | 0.066 | 0.066 | 0.125 | 0.016 | 0.016 | 0.022 | 0.014 | 0.014 | 0.019 | 0.022 | 0.022 | 0.029 |

(c) Expert policy

| | Seed 01 | | | Seed 02 | | | Seed 03 | | | Seed 04 | | | Seed 05 | | |
|---|---|---|---|---|---|---|---|---|---|---|---|---|---|---|---|
| Method | Train | Test | Bound | Train | Test | Bound | Train | Test | Bound | Train | Test | Bound | Train | Test | Bound |
| NonRec-NonInf | 0.082 | 0.077 | 0.154 | 0.077 | 0.076 | 0.151 | 0.052 | 0.056 | 0.109 | 0.016 | 0.016 | 0.031 | 0.059 | 0.066 | 0.091 |
| NonRec-Inf | 0.069 | 0.064 | 0.083 | 0.054 | 0.054 | 0.068 | 0.042 | 0.046 | 0.058 | 0.010 | 0.010 | 0.017 | 0.045 | 0.048 | 0.065 |
| Rec T=2 | 0.067 | 0.063 | 0.077 | 0.059 | 0.059 | 0.074 | 0.041 | 0.046 | 0.052 | 0.011 | 0.011 | 0.018 | 0.047 | 0.051 | 0.067 |
| Rec T=6 | 0.068 | 0.064 | 0.078 | 0.050 | 0.050 | 0.062 | 0.042 | 0.047 | 0.055 | 0.012 | 0.012 | 0.019 | 0.044 | 0.046 | 0.060 |

Table 14: Normalized train error, test error, and bound for REDQ on Meta-world's Reach hard

(a) Starter policy

| Method | Seed 01 | | | Seed 02 | | | Seed 03 | | | Seed 04 | | | Seed 05 | | |
|---|---|---|---|---|---|---|---|---|---|---|---|---|---|---|---|
| | Train | Test | Bound | Train | Test | Bound | Train | Test | Bound | Train | Test | Bound | Train | Test | Bound |
| NonRec-NonInf | 0.006 | 0.006 | 0.012 | 0.007 | 0.007 | 0.012 | 0.003 | 0.003 | 0.005 | 0.231 | 0.231 | 0.234 | 0.234 | 0.234 | 0.236 |
| NonRec-Inf | 0.005 | 0.005 | 0.008 | 0.005 | 0.005 | 0.007 | 0.002 | 0.002 | 0.004 | 0.231 | 0.231 | 0.235 | 0.234 | 0.234 | 0.237 |
| Rec T=2 | 0.005 | 0.005 | 0.008 | 0.005 | 0.005 | 0.007 | 0.002 | 0.002 | 0.004 | 0.135 | 0.135 | 0.163 | 0.227 | 0.227 | 0.232 |
| Rec T=6 | 0.005 | 0.005 | 0.007 | 0.004 | 0.004 | 0.006 | 0.002 | 0.002 | 0.003 | 0.223 | 0.223 | 0.230 | 0.231 | 0.231 | 0.235 |

(b) Intermediate policy

| Method | Seed 01 | | | Seed 02 | | | Seed 03 | | | Seed 04 | | | Seed 05 | | |
|---|---|---|---|---|---|---|---|---|---|---|---|---|---|---|---|
| | Train | Test | Bound | Train | Test | Bound | Train | Test | Bound | Train | Test | Bound | Train | Test | Bound |
| NonRec-NonInf | 0.193 | 0.193 | 0.195 | 0.005 | 0.005 | 0.009 | 0.199 | 0.199 | 0.201 | 0.183 | 0.183 | 0.186 | 0.186 | 0.186 | 0.188 |
| NonRec-Inf | 0.193 | 0.193 | 0.196 | 0.004 | 0.004 | 0.006 | 0.199 | 0.199 | 0.202 | 0.183 | 0.183 | 0.187 | 0.186 | 0.186 | 0.189 |
| Rec T=2 | 0.041 | 0.041 | 0.068 | 0.004 | 0.004 | 0.006 | 0.196 | 0.196 | 0.200 | 0.024 | 0.024 | 0.029 | 0.025 | 0.025 | 0.031 |
| Rec T=6 | 0.097 | 0.097 | 0.115 | 0.003 | 0.003 | 0.005 | 0.198 | 0.198 | 0.201 | 0.025 | 0.025 | 0.032 | 0.040 | 0.040 | 0.057 |

(c) Expert policy

| Method | Seed 01 | | | Seed 02 | | | Seed 03 | | | Seed 04 | | | Seed 05 | | |
|---|---|---|---|---|---|---|---|---|---|---|---|---|---|---|---|
| | Train | Test | Bound | Train | Test | Bound | Train | Test | Bound | Train | Test | Bound | Train | Test | Bound |
| NonRec-NonInf | 0.007 | 0.007 | 0.013 | 0.010 | 0.010 | 0.022 | 0.208 | 0.208 | 0.211 | 0.011 | 0.011 | 0.020 | 0.211 | 0.211 | 0.213 |
| NonRec-Inf | 0.006 | 0.006 | 0.009 | 0.012 | 0.012 | 0.016 | 0.208 | 0.208 | 0.211 | 0.009 | 0.009 | 0.012 | 0.211 | 0.211 | 0.214 |
| Rec T=2 | 0.005 | 0.005 | 0.007 | 0.011 | 0.011 | 0.014 | 0.203 | 0.203 | 0.208 | 0.008 | 0.008 | 0.011 | 0.207 | 0.207 | 0.211 |
| Rec T=6 | 0.005 | 0.005 | 0.007 | 0.010 | 0.010 | 0.014 | 0.172 | 0.172 | 0.182 | 0.007 | 0.007 | 0.010 | 0.209 | 0.209 | 0.213 |

Table 15: Normalized train error, test error, and bound for REDQ on Meta-world's Window open

(a) Starter policy

| Method | Seed 01 | | | Seed 02 | | | Seed 03 | | | Seed 04 | | | Seed 05 | | |
|---|---|---|---|---|---|---|---|---|---|---|---|---|---|---|---|
| | Train | Test | Bound | Train | Test | Bound | Train | Test | Bound | Train | Test | Bound | Train | Test | Bound |
| NonRec-NonInf | 0.432 | 0.435 | 0.439 | 0.004 | 0.005 | 0.007 | 0.041 | 0.042 | 0.066 | 0.425 | 0.425 | 0.432 | 0.431 | 0.432 | 0.438 |
| NonRec-Inf | 0.432 | 0.435 | 0.441 | 0.005 | 0.005 | 0.008 | 0.032 | 0.032 | 0.042 | 0.425 | 0.426 | 0.435 | 0.431 | 0.432 | 0.442 |
| Rec T=2 | 0.077 | 0.076 | 0.134 | 0.005 | 0.005 | 0.008 | 0.034 | 0.034 | 0.045 | 0.056 | 0.056 | 0.068 | 0.059 | 0.059 | 0.094 |
| Rec T=6 | 0.073 | 0.071 | 0.148 | 0.010 | 0.011 | 0.015 | 0.033 | 0.032 | 0.042 | 0.054 | 0.054 | 0.082 | 0.059 | 0.059 | 0.107 |

(b) Intermediate policy

| Method | Seed 01 | | | Seed 02 | | | Seed 03 | | | Seed 04 | | | Seed 05 | | |
|---|---|---|---|---|---|---|---|---|---|---|---|---|---|---|---|
| | Train | Test | Bound | Train | Test | Bound | Train | Test | Bound | Train | Test | Bound | Train | Test | Bound |
| NonRec-NonInf | 0.066 | 0.066 | 0.137 | 0.209 | 0.209 | 0.262 | 0.440 | 0.440 | 0.452 | 0.001 | 0.001 | 0.002 | 0.066 | 0.064 | 0.128 |
| NonRec-Inf | 0.050 | 0.050 | 0.063 | 0.345 | 0.345 | 0.378 | 0.440 | 0.440 | 0.450 | 0.001 | 0.001 | 0.003 | 0.073 | 0.075 | 0.090 |
| Rec T=2 | 0.047 | 0.047 | 0.059 | 0.049 | 0.049 | 0.059 | 0.066 | 0.066 | 0.133 | 0.001 | 0.001 | 0.003 | 0.064 | 0.064 | 0.081 |
| Rec T=6 | 0.047 | 0.047 | 0.057 | 0.091 | 0.091 | 0.147 | 0.063 | 0.063 | 0.144 | 0.002 | 0.002 | 0.004 | 0.067 | 0.066 | 0.083 |

(c) Expert policy

| Method | Seed 01 | | | Seed 02 | | | Seed 03 | | | Seed 04 | | | Seed 05 | | |
|---|---|---|---|---|---|---|---|---|---|---|---|---|---|---|---|
| | Train | Test | Bound | Train | Test | Bound | Train | Test | Bound | Train | Test | Bound | Train | Test | Bound |
| NonRec-NonInf | 0.041 | 0.042 | 0.086 | 0.001 | 0.001 | 0.003 | 0.445 | 0.445 | 0.452 | 0.001 | 0.001 | 0.002 | 0.073 | 0.073 | 0.140 |
| NonRec-Inf | 0.032 | 0.033 | 0.042 | 0.002 | 0.002 | 0.003 | 0.445 | 0.445 | 0.454 | 0.002 | 0.002 | 0.004 | 0.054 | 0.054 | 0.070 |
| Rec T=2 | 0.033 | 0.034 | 0.043 | 0.002 | 0.002 | 0.004 | 0.064 | 0.064 | 0.118 | 0.002 | 0.002 | 0.004 | 0.047 | 0.047 | 0.057 |
| Rec T=6 | 0.029 | 0.030 | 0.036 | 0.003 | 0.003 | 0.005 | 0.060 | 0.059 | 0.142 | 0.002 | 0.002 | 0.005 | 0.051 | 0.051 | 0.064 |

Table 16: Normalized train error, test error, and bound for REDQ on DM Control's Walker

(a) Starter policy

| | Seed 01 | | | Seed 02 | | | Seed 03 | | | Seed 04 | | | Seed 05 | | |
|---|---|---|---|---|---|---|---|---|---|---|---|---|---|---|---|
| Method | Train | Test | Bound | Train | Test | Bound | Train | Test | Bound | Train | Test | Bound | Train | Test | Bound |
| NonRec-NonInf | 0.094 | 0.094 | 0.156 | 0.090 | 0.089 | 0.147 | 0.066 | 0.066 | 0.112 | 0.056 | 0.056 | 0.097 | 0.102 | 0.102 | 0.160 |
| NonRec-Inf | 0.084 | 0.084 | 0.152 | 0.082 | 0.082 | 0.148 | 0.054 | 0.054 | 0.107 | 0.045 | 0.045 | 0.095 | 0.073 | 0.072 | 0.133 |
| Rec T=2 | 0.089 | 0.089 | 0.155 | 0.081 | 0.081 | 0.144 | 0.058 | 0.058 | 0.112 | 0.052 | 0.052 | 0.102 | 0.081 | 0.080 | 0.142 |
| Rec T=6 | 0.062 | 0.062 | 0.117 | 0.057 | 0.057 | 0.107 | 0.039 | 0.039 | 0.084 | 0.034 | 0.034 | 0.077 | 0.053 | 0.052 | 0.101 |

(b) Intermediate policy

| | Seed 01 | | | Seed 02 | | | Seed 03 | | | Seed 04 | | | Seed 05 | | |
|---|---|---|---|---|---|---|---|---|---|---|---|---|---|---|---|
| Method | Train | Test | Bound | Train | Test | Bound | Train | Test | Bound | Train | Test | Bound | Train | Test | Bound |
| NonRec-NonInf | 0.095 | 0.095 | 0.153 | 0.089 | 0.089 | 0.146 | 0.090 | 0.090 | 0.139 | 0.083 | 0.087 | 0.131 | 0.100 | 0.100 | 0.160 |
| NonRec-Inf | 0.083 | 0.083 | 0.142 | 0.090 | 0.090 | 0.149 | 0.065 | 0.065 | 0.118 | 0.063 | 0.066 | 0.112 | 0.074 | 0.074 | 0.136 |
| Rec T=2 | 0.082 | 0.081 | 0.140 | 0.080 | 0.080 | 0.137 | 0.064 | 0.064 | 0.115 | 0.070 | 0.074 | 0.122 | 0.089 | 0.088 | 0.152 |
| Rec T=6 | 0.062 | 0.061 | 0.108 | 0.058 | 0.058 | 0.105 | 0.046 | 0.046 | 0.087 | 0.045 | 0.049 | 0.085 | 0.056 | 0.057 | 0.105 |

(c) Expert policy

| | Seed 01 | | | Seed 02 | | | Seed 03 | | | Seed 04 | | | Seed 05 | | |
|---|---|---|---|---|---|---|---|---|---|---|---|---|---|---|---|
| Method | Train | Test | Bound | Train | Test | Bound | Train | Test | Bound | Train | Test | Bound | Train | Test | Bound |
| NonRec-NonInf | 0.105 | 0.100 | 0.164 | 0.115 | 0.111 | 0.177 | 0.086 | 0.084 | 0.133 | 0.077 | 0.077 | 0.119 | 0.078 | 0.078 | 0.125 |
| NonRec-Inf | 0.097 | 0.093 | 0.168 | 0.082 | 0.077 | 0.130 | 0.077 | 0.076 | 0.130 | 0.064 | 0.065 | 0.116 | 0.071 | 0.071 | 0.122 |
| Rec T=2 | 0.097 | 0.092 | 0.166 | 0.089 | 0.085 | 0.139 | 0.068 | 0.066 | 0.124 | 0.062 | 0.062 | 0.109 | 0.075 | 0.075 | 0.128 |
| Rec T=6 | 0.069 | 0.067 | 0.121 | 0.060 | 0.058 | 0.100 | 0.052 | 0.050 | 0.096 | 0.046 | 0.045 | 0.083 | 0.054 | 0.054 | 0.096 |

Table 17: Normalized train error, test error, and bound for REDQ on DM Control's Reacher

(a) Starter policy

| | Seed 01 | | | Seed 02 | | | Seed 03 | | | Seed 04 | | | Seed 05 | | |
|---|---|---|---|---|---|---|---|---|---|---|---|---|---|---|---|
| Method | Train | Test | Bound | Train | Test | Bound | Train | Test | Bound | Train | Test | Bound | Train | Test | Bound |
| NonRec-NonInf | 0.213 | 0.248 | 0.285 | 0.208 | 0.228 | 0.278 | 0.186 | 0.195 | 0.229 | 0.174 | 0.132 | 0.212 | 0.185 | 0.198 | 0.254 |
| NonRec-Inf | 0.201 | 0.243 | 0.282 | 0.199 | 0.216 | 0.273 | 0.188 | 0.194 | 0.273 | 0.172 | 0.134 | 0.263 | 0.165 | 0.183 | 0.242 |
| Rec T=2 | 0.209 | 0.230 | 0.283 | 0.189 | 0.212 | 0.272 | 0.182 | 0.189 | 0.264 | 0.173 | 0.148 | 0.254 | 0.166 | 0.184 | 0.234 |
| Rec T=6 | 0.183 | 0.218 | 0.257 | 0.176 | 0.200 | 0.248 | 0.178 | 0.183 | 0.254 | 0.167 | 0.138 | 0.246 | 0.149 | 0.209 | 0.210 |

(b) Intermediate policy

| | Seed 01 | | | Seed 02 | | | Seed 03 | | | Seed 04 | | | Seed 05 | | |
|---|---|---|---|---|---|---|---|---|---|---|---|---|---|---|---|
| Method | Train | Test | Bound | Train | Test | Bound | Train | Test | Bound | Train | Test | Bound | Train | Test | Bound |
| NonRec-NonInf | 0.143 | 0.129 | 0.213 | 0.103 | 0.125 | 0.162 | 0.168 | 0.198 | 0.243 | 0.262 | 0.264 | 0.327 | 0.109 | 0.103 | 0.169 |
| NonRec-Inf | 0.120 | 0.091 | 0.179 | 0.091 | 0.121 | 0.147 | 0.147 | 0.177 | 0.201 | 0.234 | 0.234 | 0.299 | 0.090 | 0.084 | 0.157 |
| Rec T=2 | 0.117 | 0.084 | 0.172 | 0.092 | 0.118 | 0.147 | 0.147 | 0.176 | 0.205 | 0.235 | 0.235 | 0.301 | 0.091 | 0.084 | 0.159 |
| Rec T=6 | 0.107 | 0.067 | 0.158 | 0.072 | 0.099 | 0.120 | 0.124 | 0.147 | 0.175 | 0.225 | 0.224 | 0.287 | 0.073 | 0.067 | 0.140 |

(c) Expert policy

| | Seed 01 | | | Seed 02 | | | Seed 03 | | | Seed 04 | | | Seed 05 | | |
|---|---|---|---|---|---|---|---|---|---|---|---|---|---|---|---|
| Method | Train | Test | Bound | Train | Test | Bound | Train | Test | Bound | Train | Test | Bound | Train | Test | Bound |
| NonRec-NonInf | 0.118 | 0.121 | 0.184 | 0.099 | 0.122 | 0.153 | 0.202 | 0.180 | 0.274 | 0.112 | 0.143 | 0.171 | 0.098 | 0.108 | 0.153 |
| NonRec-Inf | 0.102 | 0.107 | 0.177 | 0.064 | 0.099 | 0.105 | 0.196 | 0.174 | 0.254 | 0.105 | 0.148 | 0.149 | 0.097 | 0.113 | 0.162 |
| Rec T=2 | 0.104 | 0.109 | 0.172 | 0.073 | 0.107 | 0.114 | 0.197 | 0.174 | 0.251 | 0.104 | 0.142 | 0.145 | 0.090 | 0.102 | 0.143 |
| Rec T=6 | 0.114 | 0.119 | 0.229 | 0.052 | 0.117 | 0.087 | 0.182 | 0.151 | 0.236 | 0.084 | 0.123 | 0.118 | 0.072 | 0.084 | 0.119 |

Table 18: Normalized train error, test error, and bound for REDQ on DM Control's Ball-in-cup

(a) Starter policy

| Method | Seed 01 | | | Seed 02 | | | Seed 03 | | | Seed 04 | | | Seed 05 | | |
|---|---|---|---|---|---|---|---|---|---|---|---|---|---|---|---|
| | Train | Test | Bound | Train | Test | Bound | Train | Test | Bound | Train | Test | Bound | Train | Test | Bound |
| NonRec-NonInf | 0.082 | 0.082 | 0.141 | 0.090 | 0.090 | 0.146 | 0.087 | 0.087 | 0.145 | 0.081 | 0.081 | 0.134 | 0.075 | 0.075 | 0.127 |
| NonRec-Inf | 0.065 | 0.066 | 0.104 | 0.072 | 0.072 | 0.120 | 0.077 | 0.077 | 0.121 | 0.066 | 0.066 | 0.113 | 0.066 | 0.066 | 0.107 |
| Rec T=2 | 0.068 | 0.068 | 0.109 | 0.068 | 0.068 | 0.106 | 0.065 | 0.065 | 0.106 | 0.068 | 0.068 | 0.109 | 0.070 | 0.070 | 0.109 |
| Rec T=6 | 0.055 | 0.055 | 0.090 | 0.047 | 0.047 | 0.079 | 0.050 | 0.050 | 0.085 | 0.051 | 0.051 | 0.085 | 0.047 | 0.048 | 0.080 |

(b) Intermediate policy

| Method | Seed 01 | | | Seed 02 | | | Seed 03 | | | Seed 04 | | | Seed 05 | | |
|---|---|---|---|---|---|---|---|---|---|---|---|---|---|---|---|
| | Train | Test | Bound | Train | Test | Bound | Train | Test | Bound | Train | Test | Bound | Train | Test | Bound |
| NonRec-NonInf | 0.101 | 0.101 | 0.160 | 0.079 | 0.079 | 0.135 | 0.066 | 0.066 | 0.119 | 0.094 | 0.093 | 0.149 | 0.077 | 0.076 | 0.131 |
| NonRec-Inf | 0.088 | 0.087 | 0.140 | 0.068 | 0.068 | 0.108 | 0.068 | 0.068 | 0.108 | 0.076 | 0.075 | 0.119 | 0.072 | 0.071 | 0.112 |
| Rec T=2 | 0.081 | 0.079 | 0.121 | 0.068 | 0.067 | 0.106 | 0.070 | 0.070 | 0.115 | 0.082 | 0.081 | 0.128 | 0.092 | 0.090 | 0.140 |
| Rec T=6 | 0.065 | 0.064 | 0.101 | 0.049 | 0.049 | 0.080 | 0.051 | 0.051 | 0.085 | 0.052 | 0.052 | 0.085 | 0.055 | 0.055 | 0.088 |

(c) Expert policy

| Method | Seed 01 | | | Seed 02 | | | Seed 03 | | | Seed 04 | | | Seed 05 | | |
|---|---|---|---|---|---|---|---|---|---|---|---|---|---|---|---|
| | Train | Test | Bound | Train | Test | Bound | Train | Test | Bound | Train | Test | Bound | Train | Test | Bound |
| NonRec-NonInf | 0.092 | 0.092 | 0.140 | 0.074 | 0.074 | 0.125 | 0.075 | 0.076 | 0.126 | 0.070 | 0.070 | 0.124 | 0.093 | 0.093 | 0.142 |
| NonRec-Inf | 0.073 | 0.073 | 0.116 | 0.073 | 0.073 | 0.118 | 0.071 | 0.071 | 0.117 | 0.071 | 0.071 | 0.116 | 0.067 | 0.067 | 0.109 |
| Rec T=2 | 0.068 | 0.068 | 0.110 | 0.074 | 0.074 | 0.118 | 0.067 | 0.067 | 0.106 | 0.071 | 0.071 | 0.110 | 0.077 | 0.077 | 0.120 |
| Rec T=6 | 0.052 | 0.052 | 0.086 | 0.054 | 0.054 | 0.090 | 0.052 | 0.052 | 0.086 | 0.053 | 0.053 | 0.089 | 0.050 | 0.049 | 0.082 |

