# OpenReview forum: "Deep Actor-Critics with Tight Risk Certificates"
_TMLR — Rejected by TMLR_

### Review · Reviewer_E9RF · 2026-01-26

**Summary Of Contributions:**

**Summary of Contributions**

The authors present a framework for generating tight risk certificates for deep actor-critic reinforcement learning (RL) algorithms. Addressing the lack of validation schemes that fully quantify the risk of malfunction in physical systems, the paper introduces a pipeline that predicts generalization performance based on validation-time observations.

Key contributions include:

1. **Recursive PAC-Bayes for Actor-Critics:** The authors adapt the Recursive PAC-Bayes approach to deep, model-free actor-critic architectures. This involves splitting validation data to recursively build bounds on excess loss, using the predictor from the previous step as a data-informed prior.


2. **Four-Step Certification Pipeline:** A structured workflow is proposed: (1) Train the agent, (2) Collect evaluation rollouts from the frozen policy, (3) Fit posteriors via recursive training of Bayesian Neural Networks (BNNs), and (4) Construct the risk certificate.


3. **Empirical Validation:** The method is evaluated across multiple algorithms (PPO, SAC, REDQ) and diverse benchmarks (MuJoCo, DM Control, Meta-World). The authors introduce the use of the Local Reparameterization Trick (LRT) in this context to significantly improve bound tightness.



### **Key Strengths**

1. **Tightness of Certificates:** The proposed recursive approach produces risk certificates that are significantly tighter than non-recursive baselines and tight enough for practical consideration.


2. **Strong Correlation with Test Error:** Empirical results show a high linear correlation between the generated risk certificates and the actual observed test errors, particularly for expert-level policies.

3. **Technical Innovation:** The successful application of the Local Reparameterization Trick (LRT) to improve the posterior in a PAC-Bayes RL setting is a specific technical strength that enhances stability and tightness.



### **Key Weaknesses**

1. **Computational Cost:** The recursive nature of the pipeline requires training a sequence of Bayesian Neural Networks (e.g., training 6 networks for ), which increases the computational overhead compared to non-recursive baselines.



2. **Simulation-Based Validation:** While the motivation is physical deployment, the experiments are currently limited to simulated environments (MuJoCo, etc.) and have not yet been demonstrated on physical hardware

**Audience:**

Yes

**Audience Explanation:**

The findings of this paper are likely to be of significant interest to multiple sub-communities within the TMLR audience: Reinforcement Learning Researchers, AI Safety and Robotics Practitioners.

**Broader Impact Concerns:**

The submission includes a Broader Impact section that discusses the potential for these risk certificates to foster trust in agentic AI and promote the safe adoption of robotic systems in society.

**Claims And Evidence:**

Yes

**Claims Explanation:**

While I am not an expert in this specific domain, I find that the submission provides accurate and clear empirical evidence to support its central claims regarding the development of tight risk certificates for deep actor-critic algorithms.

**Requested Changes:**

1. **Verification of 'Unseen' Data Status**: The current methodology relies on collecting rollouts from a frozen policy to serve as the validation set. However, there is a concern that this does not guarantee the data is truly "unseen." If the policy has converged (especially an "Expert" policy), the new rollouts collected in the same simulation environment may be nearly identical to the trajectories observed during training. In this case, the risk certificate might effectively be measuring the policy's performance on memorized data rather than its generalization capability. **Request**: It is strongly recommended to either provide a formal proof/analysis showing that the validation dataset is statistically distinct from the training data, or to validate the risk certificates on a physical robot (or a simulation with perturbed dynamics). This is necessary to demonstrate that the certificates truly predict performance on unseen scenarios rather than just reproducing training performance.

2. **Theoretical Justification of the I.I.D. Assumption**: The validity of the "Risk Certificates" heavily relies on the assumption that the validation data is i.i.d., a prerequisite for the standard PAC-Bayes bounds used (e.g., Lemma A.1 ). The submission addresses the non-i.i.d. nature of RL trajectories by applying a "simple thinning strategy" to "better approximate the i.i.d. assumption".
While thinning reduces correlation, it does not mathematically guarantee the independence required for the rigorous probabilistic guarantees implied by the term "certificate."
**Request**: Please explicitly discuss the theoretical limitations of using thinning as a proxy for i.i.d. data in this context. Acknowledge that the "certificates" hold only insofar as this approximation holds, or cite relevant theoretical work justifying thinning for PAC-Bayes bounds in mixing processes. This is critical because the claim of providing "tight risk certificates"  implies a level of mathematical rigor that is challenged by this heuristic approximation.

3. **Clarification of "Risk of Malfunction" vs. Performance Prediction**:
The abstract frames the problem around quantifying the "risk of malfunction" in physical systems. However, the methodology focuses on bounding the "expected discounted return" (generalization performance).
**Request**: Please clarify the relationship between low discounted returns and "malfunction." In many safety-critical contexts, "risk" implies violating specific safety constraints (e.g., falling, collisions) rather than just achieving a lower reward. Precision in terminology would prevent readers from conflating performance bounds with safety/constraint-satisfaction guarantees.

4. **Sensitivity Analysis of the Recursion Parameter ($\kappa$)**:
The Recursive PAC-Bayes implementation uses a fixed scaling factor of  for all experiments.
**Request**: Please include a brief sensitivity analysis or discussion regarding this hyperparameter. Does the tightness of the bound vary significantly if  is adjusted? Understanding the robustness of the method to this parameter would strengthen the empirical evaluation.

5. **Computational Complexity and Trade-offs**:
The recursive approach ($T=6$) requires training a sequence of Bayesian Neural Networks, whereas the non-recursive baselines presumably require fewer . While the appendix lists total compute time, the main text would benefit from a direct comparison of the wall-clock time required to generate a certificate using the recursive method versus the baselines.
**Request**: Please explicitly discuss the computational cost trade-off. Is the significant gain in tightness worth the additional training time for the recursive steps?

6. **Discussion on "Minimal" Data for Physical Systems**:
The paper argues that a "small feasible set of evaluation rollouts" is sufficient and motivates the work via deployment on physical systems. The experiments use 100 episodes for validation.
**Request**: Please contextualize "100 episodes" in terms of physical robot deployment. While small for deep learning, 100 episodes can be substantial for hardware in terms of time and wear. A brief discussion on the practical feasibility of collecting 100 evaluation episodes on real hardware would better support the claim of "minimal evaluation data".

---

> ### Author Response · Authors · 2026-03-02
>
> Thank you for your thorough review.
>
> **Verification of 'Unseen' Data Status**
> While trajectories from an expert policy exhibit lower variance between them, they remain stochastic and independent of the training data in the sense required by a PAC-Bayesian bound.
> The aim of RPBRL is not to provide certificates robust to domain shift or perturbed dynamics. Instead, it certifies the expected performance of a given task and policy.
> Please note that the qualitative pattern of the various bounds relative to each other is consistent across policy levels, i.e., not due to convergence.
>
> **Theoretical Justification of the I.I.D. Assumption**
> Indeed, thinning, as commonly done in the Markov chain Monte Carlo sampling literature, only reduces the correlation between samples. We've added a discussion on this in Section 3 (ii).
>
> > As conservative samples are highly correlated, we apply a simple thinning strategy analogous to thinning in Markov chain Monte Carlo (MCMC). Subsampling of correlated samples in such chains reduces short-range autocorrelation without strictly enforcing independence. We sample by taking every $m$-th sample depending on the environment to better approximate the i.i.d.\ assumption required by PAC-Bayes and to form a data set $\mathcal D = \{(s_n, G_n)\}_{n=1}^N$.  Our certificates should be interpreted as holding under this approximation. The experimental evaluation in Section 4 demonstrates empirically that, even under this approximation, the approach provides informative and well-correlated bounds.
>
> **Clarification of "Risk of Malfunction" vs. Performance Prediction**
> Within this work we use risk in the sense of performance degradation as measured via the expected discounted return.
> Malfunction in an agent, physical or not, therefore refers only to such a degradation rather than a violation of external safety constraints.
>
>
>
> **Sensitivity Analysis of the Recursion Parameter ($\kappa$)**
> Our choice of $\kappa = 0.5$ follows the work by Wu et al. (2024), who found fixing this value to be sufficient for their experimental evaluation.
> See Section 4 in Wu et al. (2024) for a discussion of how an optimal $\kappa_t$ could be chosen for each step of the recursion $t$.
> We have added a discussion on this in Section 3 (iii):
>
> > We follow the experimental setup of Wu & Seldin (2022) and choose $\kappa_t = 0.5$ for all $t$ throughout our experiments. See Wu & Seldin (2022, Section 4) for a proposal to choose $\kappa_t$ via a grid-based approach.
>
>
> **Computational Complexity and Trade-offs**
> We agree that training a set of $T$ BNNs requires additional computational cost. Note, however, that this certification process is conducted offline and post-training. Given that these are small two-layer neural nets their training cost is negligible compared to the main computational burden of learning the policy.
>
> **Discussion on "Minimal" Data for Physical Systems**
> Our usage of _minimal_ indeed relates to deep RL agents, which often require hundreds of thousands of environmental interactions, making 100 additional episodes negligible. Note also Figure 4, which shows that if additional data collection is a constraint, even using just 25 episodes already yields strong results.
> In practice, this relates to the point above. If data collection is a bottleneck, it can be traded against computational cost by using a smaller set of episodes but a deeper recursion.

---

### Review · Reviewer_T8X7 · 2026-02-02

**Summary Of Contributions:**

This paper enables PAC-Bayes risk certificates for reinforcement learning by adopting an approach previously only tested in classification tasks. These bounds are empirically evaluated on continuous control environments using three actor-critic algorithms. The results show that the bounds have a strong correlation with the actual test error in most cases.

**Additional Comments:**

In some places, the lower part of the text is cut off/white. I'm not sure if this is a LaTex or OpenReview issue, but might be worth checking for a final version.

**Audience:**

Yes

**Audience Explanation:**

This approach is fairly general, enabling risk certification for it seems like any kind of policy, only requires moderate validation data and seems to produce accurate bounds in most experiments. This is an important combination of factors for many RL applications.

**Broader Impact Concerns:**

None. Risk certifications will make RL applications easier since we can gauge the policy impact better, but the discussion in the paper captures this sufficiently.

**Claims And Evidence:**

Yes

**Claims Explanation:**

Methodologically, the paper is quite simple as it transfers the process for computing the bounds almost entirely from previous work and can apply it to RL with minimal changes (i.e. simple data thinning). This means the evaluation can focus on how well the bounds work for test time error prediction. With three algorithms on three performance levels, three environment suits and five training seeds each run, I believe the evaluation is broad enough to support the conclusion. The ablations into the use of recusion and validation data size give important context on what to expect in practice. I also appreciate the experimental details in the appendix.

I find it quite unfortunate, however, that the DM control and MetaWorld results are mentioned in the experimental setup, but not shown or discussed in the main paper. There is certainly still page space left and while the DM control results seem quite comparable to Mujoco, MetaWorld's tightness seems quite different. In addition, the correlations for these environments are not plotted and only supplied as raw numbers. I believe the authors should discuss these domains properly in the main paper (for MetaWorld it seems like there should at least be a sentence that the bounds do not behave as nicely as on the other domains), even if this means not including all ablations. I would be quite curious, for example, to see the reason why the MetaWorld results seem to vary so much.

**Requested Changes:**

Please extend the result discussion a bit to include DM control and especially MetaWorld. To me it would be fine if this means e.g. leaving the recusion ablation out of the main paper for these domains, but still have a summary of the best seen correlation and tightness.

I also don't understand why the data thinning is not mentioned in the main text. From what I gather in the appendix, it is very simple and would only need an additional one or two sentences, so I don't know how to justify moving this significant detail (since it's part of the argument why these approaches haven't previously been applied to RL) to the appendix.

---

> ### Author Response · Authors · 2026-03-02
>
> Thank you for your detailed review.
>
> **Including DM control and MetaWorld results in the main paper.**
> Thank you for the suggestion. In our revised version, we now include them alongside the MuJoCo discussion in Section 4. *(See the updated pdf once it is available for the restructured text.)*
>
> **Different behavior of the MetaWorld bounds**
> Our assumption regarding the relative behavior of the bounds stems from the inherent complexity and variability of its environments and the noise that adds to the results.
> Figure 6 demonstrates that switching from an uninformed to an informed prior provides additional tightness, while deeper recursion does not help tighten the bounds further. We have added this to our new discussion.
>
>
> **Correlation for DMC and MetaWorld is not plotted but only supplied as raw numbers**
> Thank you for the suggestion. We've added the related plots to our updated discussion in Section 4.
>
> **Discuss thinning in the main paper.**
> Thank you for the suggestion. We have added the following in Section 3 (ii).
>
> > As conservative samples are highly correlated, we apply a simple thinning strategy analogous to thinning in Markov chain Monte Carlo (MCMC). Subsampling of correlated samples in such chains reduces short-range autocorrelation without strictly enforcing independence. We sample by taking every $m$-th sample depending on the environment to better approximate the i.i.d.\ assumption required by PAC-Bayes and to form a data set $\mathcal D = \{(s_n, G_n)\}_{n=1}^N$.  Our certificates should be interpreted as holding under this approximation. The experimental evaluation in Section 4 demonstrates empirically that, even under this approximation, the approach provides informative and well-correlated bounds.
>
>
> **Truncation of parts of the text**
> Could you clarify where you observe the truncated text? We cannot reproduce that with our local pdf readers.

---

> > ### Comment · Reviewer_T8X7 · 2026-03-12
> >
> > Thank you for these changes to the paper, I think they improve it quite a bit. With the updated version I also don't see truncated text anymore, maybe this was a local issue for me.

---

### Review · Reviewer_56AN · 2026-03-05

**Summary Of Contributions:**

This paper aims to certify the performance of learned policies in RL by providing high-confidence bounds on the quality of the policy. These bounds are developed using the PAC-Bayes framework, which has historically focused on the supervised learning setting.

While the theoretical tools used in this paper are interesting, the approach using PAC-Bayes bounds is not well-motivated in this setting as opposed to other policy evaluation methods. Additionally, clarity issues make it difficult to understand what exactly is being reported in the experiments and evaluate the proposed algorithm.

**Audience:**

Yes

**Audience Explanation:**

Yes, I think that the central problem---producing high-confidence evaluations of the performance of RL algorithms---would broadly be of interest to the community. For practictioners, robustness certificates would be helpful to ensure reliable RL agents while, for theoreticians, applying PAC-Bayes bounds in a new setting could be interesting.

**Broader Impact Concerns:**

None.

**Claims And Evidence:**

No

**Claims Explanation:**

First, there are clarity issues making it more difficult to understand the contributions and what is done in the paper:
- The framing and motivation for the proposed approach is unclear. At the start, the goal seems to be: Produce a high-confidence bound on the quality of a learned policy.
This does not seem to match the experimental metrics though. In the experiments, the reported plots compare "bound" and "test error", which is confusing since I would have expected a plot of "bound" vs. "performance of policy" where the bound indicates a lower bound on how much return the policy collects. Instead, the paper seems to learn a separate "critic" using evaluation episodes after training is done, whose error is then bounded. This is not clearly described.

- While the paper emphasizes actor-critic algorithms in the title, background and experiments sections, the proposed algorithm does not seem to rely on this fact at all. Rather, it seems to be a general approach that could work for any policy. Policies could be learned through other means or even produced without any learning at all. The emphasis on actor-critic algorithms and learned policies seems to be misleading.

Aside from the clarity, I have two major concerns about the proposed appoach:
- If the intended goal is to give a high-confidence estimate of the performance of a fixed policy, this would fit squarely in the realm of "policy evaluation". There are many works in this area and this literature is entirely missing in the paper. It would be best to also compare the proposed algorithm to at least one approach from this field e.g. "Data-Efficient Off-Policy Policy Evaluation for Reinforcement Learning" by Thomas and Brunskill. In fact, the proposed approach seems much weaker than other off-policy policy evaluation methods since it has to run additional validation episodes wheras off-policy policy evaluation algorithms can utilize all the previous data, including those from the training phase.

- The motivation for using PAC-Bayes bounds is unclear. In supervised learning, PAC-Bayes bounds are used to provide generalization guarantees: Given a dataset $D$ and some hypothesis set $H$, we can bound the error under the true data distribution for any posterior over $H$ simultaneously. This uniform bound allows us to optimize the posterior freely and still get a generalization guarantee. In the  paper-under-review's setting, the policy being evaluated is fixed. If we want to evaluate that policy, it's unclear why we need to train a separate predictor and try to bound its error rather than directly bound an estimate of the performance of the policy. For example, as a naive baseline, we could run $n$ evaluation episodes and average the returns for each episode along with a Hoeffding/Bernstein high-confidence bound.

Concerning the experiments:
- In the experiemnts, it is observed that there is a high correlation between the "bound" and the "test error". Quote from p.9: "The bounds therefore provide a reliable prediction of the test-time return, thereby answering Q1". Q1 was: "Can the test-time return of a policy be predicted with high precision?" I do not understand how we can infer that test-time predictions are accurate from high-correlation between the bound and test error. If the bound is large and this correlates with large test error, this would tell us that the estimate is, in reality, _not_ accurate.
- Minor point: It may be useful to add a x=y line on the plots to better see how the bound compares to the true values because the x-scale and y-scale don't always match.

**Requested Changes:**

For acceptance, it would be critical to address the two major concerns I have outlined above concerning the literature on policy evaluation and the motivation for the PAC-Bayes appraoach. Additionally, the clarity of the paper would have to be improved, specifically concerning how the approach is being evaluated.

---

> ### Author Response · Authors · 2026-03-06
>
> Thank you for your thorough and constructive review, we appreciate the recognition of the relevancy of the problem and the specific suggestions for improvement.
>
> **Clarification of What Is Being Bounded**
> We agree that the connection between our PAC-Bayesian bound and policy performance was not stated clearly enough. To clarify: our approach does not directly bound the policy's return. Instead, it bounds the expected squared prediction error of a Bayesian neural network (BNN) that is trained to predict discounted returns $G_t$ from states $s_t$ collected under the frozen policy. A tight bound on this prediction error certifies that the BNN accurately approximates the true value function $V^\pi$, and thereby provides a certified, state-dependent assessment of the policy's expected performance.
>
> We updated Step (iii) as follows:
>
> > (iii) Fitting posteriors via recursive training.
> The prediction target is the discounted return $G_t$ observed at
> state $s_t$. The hypothesis $h$ is a Bayesian neural network (BNN)
> whose posterior $\rho_t$ is inferred by directly minimizing the
> PAC-Bayes bound $\mathcal E_t(\rho_t, \kappa_t)$ from Theorem 2.2,
> using a bounded loss $\ell$ normalized to $[0, B]$. This
> self-certified training ensures that the bound is tight by
> construction, guaranteeing that the predictor reliably estimates
> the policy's expected discounted return from any visited state and
> thereby providing a state-dependent risk certificate for the frozen
> policy $\pi$. To obtain data-informed priors that further tighten
> the bound, the data is partitioned into $T$ disjoint subsets
> ${\mathcal D = S_1 \cup \cdots \cup S_T}$ and a sequence of $T$ BNNs is
> trained, where the $t$-th network uses $\rho_{t-1}$ as a
> data-informed prior for $t>1$ and an uninformative prior for
> $\rho_0$.
> We follow the experimental setup of Wu et al. (2024) and choose $\kappa_t = 0.5$ for all $t$ throughout our experiments. See Wu et al. (2024, Section 4) for a discussion on how to choose $\kappa_t$ via a grid-based approach.
>
> We have also revised the beginning of Section 4 to define the experimental metrics unambiguously before presenting results.
>
> > In all figures, "bound" refers to the PAC-Bayes upper bound on the expected squared return-prediction error of the BNN, and "test error" refers to the actual squared return-prediction error evaluated on held-out episodes not used during bound construction.
>
>
> **Generality Beyond Actor-Critic Algorithms**
> You are correct, the approach does not only hold for by actor-critic methods. Our emphasis on actor-critic algorithms is a scoping choice for the empirical evaluation: these methods dominate continuous control, which is the domain where risk certification is most urgently needed for safety-critical deployment. We have added a clarification in Section 3.
> > While we focus on actor-critic policies in our experiments, the certification pipeline described in steps (ii)–(iv) applies to any fixed stochastic policy $\pi$, regardless of how it was obtained.
>
>
> **Relationship to the Policy Evaluation Literature**
>
> Thank you for that suggestion. The off-policy policy evaluation (OPE) literature is indeed relevant. The key distinction is as follows: OPE methods such as importance-sampling estimators (Precup et al., 2000; Thomas & Brunskill, 2016) aim to estimate $V^\pi$ from data collected by a different behavior policy, typically by reweighting trajectories with importance ratios. Our approach instead collects fresh on-policy rollouts from the frozen policy, avoiding the high variance inherent in importance weighting. This makes the two settings complementary rather than directly competing.
>
> We have added the following to Related Work in Section B:
>
> > Off-policy policy evaluation (OPE) methods (Precup et al., 2000; Thomas & Brunskill, 2016; Jiang & Li, 2016; Dudík et al., 2014) estimate the value of a target policy from data collected under a different behavior policy, typically using importance sampling. Our setting differs in that we collect fresh on-policy rollouts from the frozen policy, which avoids the high variance of importance-weighted estimators. OPE is advantageous when additional data collection is infeasible, whereas our approach is designed for settings where a modest number of evaluation episodes can be collected, a realistic assumption for the robotic deployment scenarios we target. The two approaches are complementary: OPE methods could be used to reuse training data, while our PAC-Bayesian certificates provide formally guaranteed bounds on a learned value predictor.

---

> > ### Author Response · Authors · 2026-03-06
> >
> > **Motivation for PAC-Bayesian Bounds**
> > We would like to clarify the difference between PAC bounds, which are uniform across a hypothesis space, and PAC *"Bayes"* bounds, which are calculated with reference to a prior distribution that determines the preference weight of each hypothesis. When a higher preference weight is assigned to the true hypothesis, the bound is tighter. In a standard supervised learning application, PAC Bayes is used to predict the generalization error of a posterior-weighted hypothesis space (see, e.g., Alquier, 2024; Dziugaite & Roy, 2017). The goal could be to decide whether to deploy the expectation of this posterior to a target plant. The bound may be built *on top of* an existing posterior, or the posterior itself may be optimized with respect to the bound. The second approach, called *self-certified* learning, is an option but not an obligation to build useful PAC Bayes bounds that give meaning risk certificates. In fact, self-certified learning is a last resort, as it often restricts the capacity of a learning algorithm solely for the sake of the generalization error prediction step, which can also be performed in a post hoc manner. We choose the post hoc certification approach to make our solution agnostic to the algorithm used to train the policy for which the risk certificate will be given. The bound we develop is on the generalization performance of a "policy evaluation network," which gives a performance guarantee on the generalization performance of a policy learned by a certain RL algorithm. The generalization performance corresponds to the return of the learned policy on evaluation episodes. We do not see a difference between our means of employing PAC Bayes and its standard applications in supervised learning. Uniform bounds are by definition not PAC Bayes, and they are largely inapplicable to certification due to the intractability of the complexity term, e.g., VC dimension, Rademacher complexity, or Kolmogorov complexity. Their application even to deep neural networks in plain supervised learning setups is impractical, oftentimes impossible. This fact can be verified from the absence of non-vacuous risk certificates built on uniform PAC bounds in the risk certification literature. Therefore, we do not consider such a comparison within the scope of our paper, as it does not contribute to our key message.
> >
> > We've added the following clarification at the end of Section 3.
> > > Why a learned value predictor? The tightness of PAC-Bayes bounds is governed by the KL divergence between posterior and prior: the closer the prior is to the true solution, the tighter the bound. Achieving tight certificates therefore requires data-informed priors, i.e., priors that already approximate the value function well before the bound is evaluated. This, in turn, necessitates a parametric model whose prior can be shaped by data. We use a Bayesian neural network as this model: a portion of evaluation data is used to train an informed prior, and the remaining data is used to certify the posterior's generalization error via the PAC-Bayes bound. The bound is tight precisely because the data-informed prior mechanism keeps the KL divergence small.
> >
> > **Correlation Does Not Imply Accurate Prediction**
> >
> > Indeed, high correlation between bounds and test errors does not by itself demonstrate that the bounds are tight. Our answer to Q1 rests on *both* correlation and tightness. The tightness of the bounds (Bound - Test Error) is reported in Figures 14–18 in the appendix, where the recursive bounds with $T=6$ become tighter as the recursion increases.
> > We've shifted them to the main text.
> > Additionally, have added $x = y$ reference lines to all scatter plots (Figures 2–4, 12–13) so that the reader can immediately see whether and by how much the bound overestimates the true error. Since the bound is an upper bound, all points are expected to fall below the $x = y$ line. The distance to the line indicates tightness.
> >
> > ---
> >
> > - Alquier (2024), *User-friendly introduction to PAC-Bayes bounds*
> > - Dudík et al. (2014), *Doubly robust policy evaluation and optimization*
> > - Dziugaite & Roy (2017), *Computing non-vacuous generalization bounds for deep (stochastic) neural networks with many more parameters than training data*
> > - Jiang & Li (2016), *Doubly robust off-policy value evaluation for reinforcement learning.*
> > - Precup et al. (2000), *Eligibility traces for off-policy policy evaluation*
> > - Thomas & Brunskill (2016), *Data-efficient off-policy policy evaluation for reinforcement learning.*

---

### Decision · Action_Editor_4pE2 · 2026-05-19

**Recommendation:** Reject

**Audience:**

Yes

**Audience Explanation:**

All reviewers agree that this paper studies a relevant problem to the RL community, also that the technique adopted in the paper is of interest on its own. Therefore the AE does believe that this could be an interesting work with its weaknesses fully addressed.

**Claims And Evidence:**

No

**Claims Explanation:**

While two out of three reviewers find this paper interesting and vote favorably in acceptance, there are major concerns raised by reviewer 56AN that need to be addressed. Specifically, "My main issue is that the bounds provided by the proposed algorithm are on the error of the critic's prediction, not on the quality of the value of the policy itself. Thus, if the true prediction error is large, then the proposed algorithm cannot assess the the value of the policy. It only tells us that the prediction is in fact inaccurate" and "In particular, the policy evaluation literature provides many algorithms that do provide a high-confidence bound on the policy's performance. These can be used with past off-policy data but also new on-policy data if that is available.". The AE does agree that these are fundamental limitations of the current version which need to be addressed before being accepted.

**Resubmission Of Major Revision:**

The authors may consider submitting a major revision at a later time.